REGISTERED REPORT

# Registered report: Coding-independent regulation of the tumor suppressor PTEN by competing endogenous mRNAs

**Mitch Phelps[1], Chris Coss[1], Hongyan Wang[1], Matthew Cook[2], Reproducibility Project: Cancer Biology\***

[1]Pharmacoanalytic Shared Resource, The Ohio State University, Columbus, United States; [2]University of California San Francisco, San Francisco, United States

**Abstract** The Reproducibility Project: Cancer Biology seeks to address growing concerns about reproducibility in scientific research by conducting replications of selected experiments from a number of high-profile papers in the field of cancer biology. The papers, which were published between 2010 and 2012, were selected on the basis of citations and Altmetric scores (*Errington et al., 2014*). This Registered Report describes the proposed replication plan of key experiments from "Coding-Independent Regulation of the Tumor Suppressor PTEN by Competing Endogenous 'mRNAs' by Tay and colleagues, published in *Cell* in 2011 (*Tay et al., 2011*). The experiments to be replicated are those reported in Figures 3C, 3D, 3G, 3H, 5A and 5B, and in Supplemental Figures 3A and B. Tay and colleagues proposed a new regulatory mechanism based on competing endogenous RNAs (ceRNAs), which regulate target genes by competitive binding of shared microRNAs. They test their model by identifying and confirming ceRNAs that target *PTEN*. In Figure 3A and B, they report that perturbing expression of putative *PTEN* ceRNAs affects expression of PTEN. This effect is dependent on functional microRNA machinery (Figure 3G and H), and affects the pathway downstream of PTEN itself (Figures 5A and B). The Reproducibility Project: Cancer Biology is a collaboration between the Center for Open Science and Science Exchange, and the results of the replications will be published by *eLife*.

**\*For correspondence:** nicole@scienceexchange.com

**Group author details:**
Reproducibility Project: Cancer Biology See page 33

## Introduction

microRNAs are one of the first identified classes of non-coding RNAs that can modulate the expression of mRNA-coding transcripts by binding to complementary regions in a target gene's sequence and repressing its expression. Thus, expression levels and availability of these microRNAs can influence gene expression, and there is growing evidence that misregulation of microRNAs is correlated with some forms of cancer (*Sen et al., 2014*). Naturally occurring microRNA 'sponges' have been shown to be effective in regulating gene expression by altering the levels of their cognate microRNAs (*Choi et al., 2007*; *Karreth and Pandolfi 2013*). Poliseno and colleagues proposed that pseudogenes, long non-coding RNAs with strong homology to coding sequences, could act as the modulators of gene expression as microRNA sponges (*Poliseno et al., 2010*). They demonstrated that the pseudogene *PTENP1* could regulate the expression levels of *PTEN* via their cognate microRNAs miR-19b and miR-20a (*Poliseno et al., 2010*).

In this study, Tay and colleagues expanded upon the previous work to propose a unifying hypothesis of regulatory networks composed of competing endogenous RNAs (ceRNAs) (*Karreth and Pandolfi 2013*; *Sen et al., 2014*; *Kartha and Subramanian, 2014*). They suggest that protein-coding RNAs, not just non-coding RNAs, can cross-regulate each other based on competition for shared microRNA regulators; ceRNAs can titrate microRNAs from their target genes (*Tay et al., 2011*).

Continuing their focus on the regulation of *PTEN*, one of the most frequently mutated genes in cancer (*Song et al., 2012*), Tay and colleagues propose a computational model to identify ceRNAs de novo, termed MuTaME. Using MuTaME, they identified potential ceRNA regulators of *PTEN*, and validated if these candidate ceRNAs could modulate *PTEN* expression in a microRNA-dependent manner (*Tay et al., 2011*).

In Figure 3C, Tay and colleagues examine if silencing ceRNAs targeting *PTEN* would affect the expression levels of a luciferase construct carrying the 3'UTR of *PTEN*. They co-transfected DU145 cells with siRNAs against the candidate *PTEN* ceRNAs along with a luciferase-*PTEN 3'UTR* construct and measured luciferase activity. After confirming knockdown of each target ceRNA (Supplemental Figure 3A), they reported that the loss of three of their candidate ceRNAs - *SERINC1*, *VAPA* and *CNOT6L*, but not *ZNF460* - reduced the luciferase activity of the *PTEN 3'UTR* construct. This experiment will be replicated in Protocol 1.

In Figure 3D, they demonstrated that only the 3'UTRs of the candidate ceRNAs were required to affect changes in the luciferase activity of the *PTEN* 3'UTR construct. Ectopic overexpression of the 3'UTRs of the three candidate ceRNAs relieved inhibition of the *PTEN* 3'UTR, as evidenced by increased luciferase activity as compared to controls. This experiment will be replicated in Protocol 2.

To test if this effect was dependent on microRNAs, Tay and colleagues repeated these experiments in *DICER1* mutant HCT116 cells, in which the machinery required for microRNA function is abrogated. Transfection of wild type HCT116 cells with siRNAs targeting the candidate ceRNAs showed a marked reduction in PTEN protein levels, an effect that was not seen in the *DICER1* mutant HCT116 cells (Figures 3G and H). Knockdown of the candidate ceRNAs was confirmed by RT-PCR (Supplemental Figure 3B). This experiment will be replicated in Protocol 3.

PTEN negatively regulates the PI3K/AKT pathway (*Stambolic et al., 1998*), so Tay and colleagues examined if ceRNA modulation affected the phosphorylation of AKT. Loss of *CNOT6L* and *VAPA* in DU145 cells elevated pAKT levels after serum stimulation, an effect that was also observed in wild-type HCT116 cells (Figure 5A). However, this effect was abrogated in *DICER1* mutant HCT116 cells (Figure 5A). They also examined the effect of ceRNAs on the tumorigenic properties conferred by loss of PTEN. Silencing of the ceRNAs *CNOT6L* and *VAPA* increased cell proliferation of DU145 cells and wild-type HCT116 cells, similar to silencing of *PTEN* directly (Figure 5B). This effect was less pronounced in *DICER1* mutant HCT116 cells (Figure 5B). These experiments will be replicated in Protocol 4 and 5.

Two papers published simultaneously provide support for the actions of ceRNA regulatory networks. Karreth and colleagues, from the same lab as Tay and colleagues, demonstrated in vivo evidence for the actions of ceRNA regulation using the sleeping beauty transposase system in a mouse model of melanoma to identify and confirm putative *PTEN* ceRNAs (*Karreth et al., 2011*). Karreth and colleagues identified *CNOT6L* as a putative *PTEN* ceRNA through the sleeping beauty transposase system, providing further evidence that *CNOT6L* is indeed involved in *PTEN* regulation. Karreth and colleagues focused on *ZEB2*; using siRNA silencing, they reported that the loss of *ZEB2* reduced PTEN protein levels, and affected downstream phosphorylation of AKT (*Karreth et al., 2011*). As seen in Tay and colleagues, these effects were dependent on functional microRNA processing; *ZEB2* depletion did not affect PTEN levels in *DICER1* mutant HCT116 cells (*Karreth et al., 2011*). Sumazin and colleagues used a bioinformatics approach to identify post-translational regulation and elucidated over 7,000 genes they proposed acted as miRNA sponges. By comparing the miRNA programs of genes, they could identify genes with common miR programs, indicating the potential for miRNA-mediated crosstalk between those two genes (*Sumazin et al., 2011*). They tested their findings by exploring the regulation of *PTEN*, demonstrating that silencing of putative miRNA program-mediated regulators (mPRs) of *PTEN* decreased *PTEN* expression, and, conversely, that the perturbation of PTEN levels could inversely affect the expression of its mPRs. These manipulations also affected tumor cell growth rates, indicating potential in vivo effects of changes to mPR regulatory networks (*Sumazin et al., 2011*). Since the publication of these three papers, numerous other examples of ceRNA regulation have been reported in muscle differentiation (*Cesana et al., 2011*), human embryonic stem cell renewal (*Wang et al., 2013*), regulation of sex determination by SRY (*Granados-Riveron and Aquino-Jarquin 2014*), breast cancer (*Yang et al., 2014*; *Zheng et al., 2015a*; *2015b*), lymphoma (*Karreth et al., 2015*) and the regulation of the tumor-related HMGA1 (*Esposito et al., 2014*).

The Pandolfi group followed up on their 2011 paper by generating a mathematical model to predict optimal conditions for ceRNA activity, based on a molecular titration mechanism whose effects were correlated to the relative levels of the ceRNA and its target (*Ala et al., 2013*). They then tested their in silico predictions by experimentally exploring the effect of *VAPA* on *PTEN* expression. While silencing of *VAPA* did decrease *PTEN* expression in all five cell lines tested, they noted that the amount of silencing was correlated with the initial *VAPA:PTEN* expression ratio (*Ala et al., 2013*). However, Denzler and colleagues challenge the notion that perturbations in ceRNA expression levels could affect target genes at all (*Denzler et al., 2014*). Denzler and colleagues and Ala and colleagues both state that ceRNA effects are dependent on the kinetics of binding, which in turn relies upon the ratio of microRNAs to target sites; increasing the number of target sites through expression of ceRNAs is postulated to affect target gene repression. By quantifying the absolute copy number of the well-studied highly abundant miR-122 and its target sites, Denzler and colleagues showed that large, physiologically unlikely changes in ceRNA expression levels would be required to alter the microRNA: target site ratio enough to perturb target gene expression, casting doubt on the ability of these putative ceRNAs to affect changes in target gene expression levels (*Broderick and Zamore 2014*; *Denzler et al., 2014*). This view was contradicted by Bosson and colleagues, who identified over 3,000 high affinity target sites they claimed could be affected by ceRNAs due to low endogenous microRNA: target site ratios (*Bosson et al., 2014*). The activity and impact of potential ceRNA networks is an area of active interest (for review, see *de Giorgio et al., 2013*).

## Materials and methods

Unless otherwise noted, all protocol information was derived from the original paper, references from the original paper, or information obtained directly from the authors. An asterisk (*) indicates data or information provided by the Reproducibility Project: Cancer Biology core team. A hashtag (#) indicates information provided by the replicating lab.

### Protocol 1: Knock-down of ceRNA network genes results in decreased PTEN-3'UTR luciferase expression

This protocol describes how to silence expression of ceRNA network genes and measure effects on *PTEN* expression by measuring *PTEN* 3'UTR luciferase activity, as seen in Figures 3C and Supplementary S3A.

### Sampling

- This experiment will include four biological replicates (Luciferase assay) and four biological replicates (qRT-PCR) for a minimum power of 80%.
  - See Power Calculations section for details.
- Each experiment consists of DU145 cells co-transfected with a luciferase-*PTEN* 3'UTR reporter construct and siRNA against *PTEN* ceRNAs:
  - Cohort 1: siRNA against nontargeting control 2 (siNC)
  - Cohort 2: siGENOME siRNA against *SERINC1* (siSER)
  - Cohort 3: siGENOME siRNA against *ZNF460* (siZNF)
  - Cohort 4: siGENOME siRNA against *VAPA* (siVAPA)
  - Cohort 5: siGENOME siRNA against *CNOT6L* (siCNO)
  - Cohort 6: siGENOME siRNA against *PTEN* (siPTEN)
  - Cohort 7: siGLO RISC-free siRNA (transfection control)
- Effects of silencing ceRNAs will be tested with
  - Luciferase assay of *PTEN* 3'UTR expression (Figure 3C)
  - qRT-PCR to confirm target gene silencing (Supplementary Fig S3A)
  - siGLO fluorescence to confirm transfection efficiency

### Materials and reagents

| Reagent | Type | Manufacturer | Catalog # | Comments |
|---|---|---|---|---|
| DU145 human prostate cancer cells | Cells | ATCC | HTB-81 | |
| psiCHECK-2-PTEN 3'UTR plasmid | Plasmid | Addgene | plasmid #50936 | Communicated by original authors |
| siGLO RISC-free siRNA siGLO | siRNA | Dharmacon | D-001600-01-05 | Catalog # communicated by original authors |
| siGenome siRNA for nontargeting control 2 | siRNA | Dharmacon | D-001210-02-05 | Catalog # communicated by original authors |
| siGenome siRNA for *SERINC1* | siRNA | Dharmacon | M-010725-00-0005 | Catalog # communicated by original authors |
| siGenome siRNA for *ZNF460* | siRNA | Dharmacon | M-032012-01-0005 | Catalog # communicated by original authors |
| siGenome siRNA for *VAPA* | siRNA | Dharmacon | M-021382-01-0005 | Catalog # communicated by original authors |
| siGenome siRNA for *CNOT6L* | siRNA | Dharmacon | M-016411-01-0005 | Catalog # communicated by original authors |
| siGenome siRNA for *PTEN* | siRNA | Dharmacon | M-003023-02-0005 | Catalog # communicated by original authors |
| Dulbecco's Modified Eagle's Medium (DMEM) | Cell Culture Reagent | Invitrogen | 10313-039 | Catalog # communicated by original authors |
| Fetal Bovine Serum (FBS) | Cell Culture Reagent | Invitrogen | 10438-026 | Catalog # communicated by original authors |
| Penicillin/Streptomycin | Cell Culture Reagent | Life Technologies | 15140-163 | Communicated by original authors |
| Glutamine | Cell Culture Reagent | Life Technologies | 25030-081 | Communicated by original authors |
| Lipofectamine 2000 | Transfection Reagent | Life Technologies | 11668500 | Communicated by original authors |
| Trypsin | Transfection Reagent | Life Technologies | 15400-054 | Communicated by original authors |
| Dual Luciferase Reporter Assay | Luciferase Assay | Promega | E1960 | Catalog # communicated by original authors |
| Lysis Buffer (included with Dual-Luciferase Reporter Assay) | Buffer | Promega | E1960 | Original not specified |
| GLOMAX 96 Microplate Luminometer | Equipment | Promega | E6501 | Replaces Promega E8032 (communicated by original authors) |
| TRIzol reagent | qPCR reagent | Life Technologies | 15596026 | Communicated by original authors |
| RNeasy kit | qPCR reagent | Qiagen | 74104 | Communicated by original authors |
| High Capacity cDNA Archive kit | qPCR reagent | Life Technologies | 4368814 | Communicated by original authors |
| TaqMan probe *PTEN* | qPCR probes | Life Technologies | Hs02621230_s1 | |
| TaqMan probe *CNOT6L* | qPCR probes | Life Technologies | Hs00375913_m1 | |
| TaqMan probe *VAPA* | qPCR probes | Life Technologies | Hs00427749_m1 | |
| TaqMan probe *SERINC1* | qPCR probes | Life Technologies | Hs00380375_m1 | |
| TaqMan probe *ZNF460* | qPCR probes | Life Technologies | Hs01104252_m1 | |
| TaqMan control probe *ß-ACTIN* | qPCR probes | Life Technologies | Hs00969077_m1 | Communicated by original authors |
| TaqMan Fast Advanced Master Mix | qPCR reagent | Life Technologies | 4444964 | Communicated by original authors |
| StepOne Plus Real-Time PCR system | Equipment | Applied Biosystems | | Replaces LightCycler 480 System |
| Nanodrop 2000C Spectrometer | Equipment | Thermo Scientific | | |

## Procedure

Notes:

- All cells will be sent for mycoplasma testing and STR profiling.
- DU145 cells are maintained in DMEM supplemented with 10% FBS, #100 U/ml penicillin/100 µg/ml streptomycin, and #2 mM glutamine at 37°C in 5% $CO_2$ in a humidified atmosphere.

1. Co-transfect DU145 cells with *PTEN* 3'UTR and siRNAs:
   a. Split DU145 cells into four different cultures.
      i. These will be biological replicates.
   b. Seed cells at $1.2 \times 10^5$ cells per well in 12 well dishes for 24.
      i. Seed 13 wells: 6 transfection conditions x 2 wells per condition (for Steps 2 and 3) and 1 transfection condition (siGlo RISC free siRNA transfection control) x 1 well.
   c. Prepare separate transfection mixtures for each biological replicate.
   d. Add 100 ng of psiCHECK-2+PTEN3'UTR and 100 pmol of siRNA QS to 100 µl of Opti-MEM.
      i. Transfect a pair of wells with each of the following:
         1. siSERINC1
         2. siZNF460
         3. siVAPA
         4. siCNOT6L
         5. siPTEN
         6. non-targeting control (NC)
      ii. Transfect a single well with the following:
         1. siGLO control siRNA
   e. In a separate tube mix 2 µl of Lipofectamine 2000 with 100 µl of Opti-MEM.
      i. Scale the volume according to number of replicates.
      ii. Incubate for 10 min.
   f. Combine the plasmid/siRNA and Lipofectamine mixes with gentle mixing and incubate for an additional 20min.
   g. Aliquot 200 µl of the plasmid/siRNA and Lipofectamine transfection mix into appropriate well.
      i. Mix gently and incubate at 37°C.
      ii. Replace growth medium after 4.
      iii. After 24-48, count the number of fluorescent cells transfected with siGLO relative to total to confirm >90% transfection efficiency.
         1. If transfection is less than 90%, record efficiency, exclude replicate and omit it from the rest of the procedure. Repeat procedure until >90% efficiency is obtained.
         2. If modification to transfection is needed, record and maintain modified steps for remaining replicates.
   h. Incubate for 72 at 37°C in 5% $CO_2$ in a humidified atmosphere
2. Use one well for each transfection to measure luciferase activity:
   a. Wash cells with ice-cold PBS, aspirate and add 100 µl of 1X lysis buffer.
   b. Place on an orbital shaker for 10min to dissociate the cell layer.
   c. Pipette gently to mix and transfer 20 µl of each lysate into one well of a white-walled 96 well plate.
   d. Measure firefly and Renilla luciferase activities with the dual-luciferase reporter system with a luminometer according to the manufacturer's instructions.
3. Using the other well for each transfection, confirm siRNA target knock-down with qRT-PCR:
   a. Extract total RNA using TRIzol reagent according to manufacturer's instructions.
   b. Purify samples with RNeasy kit according to manufacturer's instructions.
   c. Quality check RNA by measuring $A_{260/280}$ and $A_{260/230}$ absorbance ratios.
   d. Total RNA can be frozen here until all biological replicates are performed after which the remaining steps will be conducted at one time.
   e. Reverse transcribe 1 µg total RNA using High Capacity cDNA Archive kit according to manufacturer's instructions.

 f. Perform qRT-PCR to confirm mRNA expression knockdown. Measure mRNA expression for each siRNA transfection sample with its appropriate target and *ß-ACTIN*, and test each probe separately using RNA from the NC transfection.
- *PTEN*
- *CNOT6L*
- *VAPA*
- *ZNF460*
- *SERINC1*
- *β-ACTIN* [endogenous control communicated by original author]
  - Prepare 10 μl real-time PCR reaction in triplicate for each reaction consisting of:
- 5 μl TaqMan mastermix
- 0.5 μl TaqMan probe for the gene of interest
- 4.5 μl cDNA (diluted 10x)
- Use standard TaqMan cycling protocol:
  1. 50°C 2 min
  2. 95°C 20 s
  3. 40 cycles of 95°C 1 s, 60°C 20 s

 g. Normalize each mRNA expression to *ß-ACTIN* and then normalize each siRNA to siNC for that transcript.

4. Repeat steps 1-3, 3 additional times

## Deliverables

- Data to be collected:
  - QC image data confirming transfection efficiency by measuring the number of fluorescent cells transfected with siGLO
  - Raw data of Renilla and firefly luciferase measures and a graph of luciferase activity for each cohort
  - QC data for total RNA ($A_{260/280}$ and $A_{260/230}$ absorbance ratios)
    qRT-PCR data to confirm silencing: raw qPCR data and for each sample and a graph of each target gene normalized with *β-ACTIN* and normalized relative to NC expression

## Confirmatory analysis plan

- Statistical Analysis of the Replication Data:
  Note: At the time of analysis, we will perform the Shapiro-Wilk test and generate a quantile-quantile plot to assess the normality of the data. We will also perform Levene's test to assess homoscedasticity. If the data appear skewed, we will perform the appropriate transformation to proceed with the proposed statistical analysis. If this is not possible, we will perform the equivalent non-parametric Wilcoxon-Mann-Whitney test.
  - Luciferase assay: One-way ANOVA of luciferase activity in DU145 cells transfected with siRNA against NC, *SERINC1*, *ZNF460*, *VAPA*, *CNOT6L*, or *PTEN*, with the following Bonferroni-corrected comparisons:
    - Non-coding siRNA vs. each of the ceRNA transfected cells (5 comparisons total).
  - qRT-PCR: Bonferroni corrected one-sample *t*-tests of normalized mRNA expression in DU145 cells transfected with siRNA against *SERINC1*, *ZNF460*, *VAPA*, *CNOT6L*, or *PTEN* compared to a constant (siNC = 1) (5 comparisons total).
- Meta-analysis of original and replication attempt effect sizes:
  - This replication attempt will perform the statistical analysis listed above, compute the effects sizes, compare them against the reported effect size in the original paper, and use a meta-analytic approach to combine the original and replication effects, which will be presented as a forest plot.

## Known differences from the original study

All known differences are listed in the materials and reagents section above with the originally used item listed in the comments section. All differences have the same capabilities as the original and are not expected to alter the experimental design.

## Provisions for quality control

Extracted RNA integrity will be reported with $A_{260/280}$ and $A_{260/230}$ absorbance ratios, and transfection efficiency will be checked using the siGLO control. qRT-PCR will be performed to confirm the silencing of ceRNA expression. The cells will be sent for mycoplasma testing confirming lack of contamination and STR profiling confirming cell line authenticity. Transfection efficiency will be recorded for each replicate and any transfection that does not contain >90% efficiency will be excluded and not continue through the rest of the procedure. Any modifications to the transfection protocol will be recorded, and the procedure will be maintained for the remaining replicates. All data obtained from the experiment - raw data, data analysis, control data and quality control data - will be made publicly available, either in the published manuscript or as an open access dataset available on the Open Science Framework (https://osf.io/oblj1/).

## Protocol 2: Overexpression of PTEN ceRNA 3'UTRs network genes results in upregulation of PTEN3'UTR luciferase activity

This protocol describes how to measure the effect of ectopic overexpression of PTEN ceRNA 3'UTRs in DU145 cells on Luc-PTEN 3'UTR levels. This protocol replicates Figures 3D.

### Sampling

- This experiment will include six biological replicates for a minimum power of 88%.
  - See Power calculations for details.
- Each experiment consists of DU145 cells co-transfected with a luciferase-PTEN 3'UTR reporter construct and:
  - Cohort 1: *SERINC1* 3'UTR (SER 3'U)
  - Cohort 2: *VAPA* 3'UTR1 (VAPA 3'U1)
  - Cohort 3: *VAPA* 3'UTR2 (VAPA 3'U2)
  - Cohort 4: *CNOT6L* 3'UTR1 (CNO 3'U1)
  - Cohort 5: *CNOT6L* 3'UTR2 (CNO 3'U2)
  - Cohort 6: *PTEN* 3'UTR (PTEN 3'U)
  - Cohort 7: Empty vector control
- Effects of overexpressing ceRNAs will be tested with
  - Luciferase assay of PTEN 3'UTR expression (Figure 3D)

### Materials and reagents

| Reagent | Type | Manufacturer | Catalog # | Comments |
|---|---|---|---|---|
| DU145 human prostate cancer cells | Cells | ATCC | HTB-81 | |
| psiCHECK-2-PTEN 3'UTR plasmid | Plasmid | Addgene | plasmid #50936 | Communicated by original authors |
| psiCHECK-2 empty vector | Plasmid | Promega | C8021 | Catalog # communicated by original authors |
| Dulbecco's Modified Eagle's Medium (DMEM) | Cell Culture Reagent | Invitrogen | 10313-039 | Catalog # communicated by original authors |
| Fetal Bovine Serum (FBS) | Cell Culture Reagent | Invitrogen | 10438-026 | Catalog # communicated by original authors |
| Penicillin/Streptomycin | Cell Culture Reagent | Life Technologies | 15140-163 | Communicated by original authors |
| Glutamine | Cell Culture Reagent | Life Technologies | 25030-081 | Communicated by original authors |
| Lipofectamine 2000 | Transfection Reagent | Life Technologies | 11668500 | Communicated by original authors |
| *SERINC1* 3'UTR vector | Plasmid | Provided by original authors | | |
| *VAPA* 3'UTR1 vector | Plasmid | Provided by original authors | | |
| *VAPA* 3'UTR2 vector | Plasmid | Provided by original authors | | |
| *CNOT6L* 3'UTR1 vector | Plasmid | Provided by original authors | | |
| *CNOT6L* 3'UTR2 vector | Plasmid | Provided by original authors | | |
| *PTEN* 3'UTR vector | Plasmid | Provided by original authors | | |
| Trypsin | Transfection Reagent | Life Technologies | 15400-054 | Communicated by original authors |
| Dual Luciferase Reporter Assay | Luciferase Assay | Promega | E1960 | Catalog # communicated by original authors |
| Luminometer | Equipment | Promega | E8032 | Catalog # communicated by original authors |

## Procedure

Notes:

- All cells will be sent for mycoplasma testing and STR profiling.
- DU145 cells are maintained in DMEM supplemented with 10% FBS, [#]100 U/ml penicillin/100 $\mu$g/ml streptomycin, and [#]2 mM glutamine at 37˚C in 5% $CO_2$ in a humidified atmosphere.

1. Transfect DU145 cells with *PTEN* 3'UTR and ceRNA 3'UTRs:
    a. Separate DU145 cells into 6 different cultures.
        i. These will be biological replicates.
    b. Seed cells at 1.2 x 10$^5$ cells per well in 12 well dishes and incubate for 24 hr.
        i. Seed 1 well per biological replicate: 7 transfections x 6 replicates.
            1. 42 wells total seeded.
    c. Prepare the transfection mix by adding 100 ng of psiCHECK-2+PTEN3'UTR and 1 μg of 3'UTR plasmid to 100 μl of Opti-MEM.
        i. Transfect one well per replicate with each of the following:
            1. *SER* 3'U
            2. *VAPA* 3'U1
            3. *VAPA* 3'U2
            4. *CNO* 3'U1
            5. *CNO* 3'U2
            6. *PTEN* 3'U
            7. empty vector control
    d. In a separate tube, mix 2 μl of Lipofectamine 2000 with 100 μl of Opti-MEM.
        i. Scale the volume of reagents accordingly.
        ii. Incubate for 10 min.
    e. Combine the plasmid and Lipofectamine mixes and incubate for an additional 20 min.
    f. Aliquot 200 μl of the plasmid and Lipofectamine transfection mix into each well. Mix gently and incubate at 37˚C in 5% $CO_2$ in a humidified atmosphere.
        i. Replace growth medium after 4 hr.
    g. Incubate for 72 hr.
2. Measure renilla and firefly luciferase activity as outlined in Protocol 1 Step 2.

## Deliverables

- Data to be collected:
    - Raw data of Renilla and firefly luciferase measures and a graph of luciferase activity for each cohort.

## Confirmatory analysis plan

- Statistical Analysis of the Replication Data:
  Note: At the time of analysis, we will test for normality and homoscedasticity of the data. If the data appears skewed, we will perform the appropriate transformation to proceed with the proposed statistical analysis. If this is not possible, we will perform the equivalent non-parametric Wilcoxon-Mann-Whitney test.
    - One-way ANOVA of luciferase activity in DU145 cells expressing 3'UTRs *SER, VAPA* 3'U1, *VAPA* 3'U2, *CNO* 3'U1, *CNO* 3'U2, *PTEN*, or empty vector control with the following Bonferroni-corrected planned comparisons:
        - Luciferase activity in each 3'UTR transfection vs. the empty vector control (6 comparisons total).
- Meta-analysis of original and replication attempt effect sizes:
    - This replication attempt will perform the statistical analysis listed above, compute the effects sizes, compare them against the reported effect size in the original paper and use a meta-analytic approach to combine the original and replication effects, which will be presented as a forest plot.

## Known differences from the original study

All known differences are listed in the materials and reagents section above with the originally used item listed in the comments section. All differences have the same capabilities as the original and are not expected to alter the experimental design.

## Provisions for quality control

The cells will be sent for mycoplasma testing confirming lack of contamination and STR profiling confirming cell line authenticity. All data obtained from the experiment - raw data, data analysis, control data and quality control data - will be made publicly available, either in the published manuscript or as an open access dataset available on the Open Science Framework (https://osf.io/oblj1/).

## Protocol 3: Knock-down of ceRNA network genes results in decreased PTEN protein that is dependent on microRNA functioning

This protocol describes how to test the effects of siRNA-mediated depletion of *SERINC1, VAPA,* or *CNOT6L* expression on PTEN protein expression in wild-type HCT116 colon cancer cells. It also tests whether these effects are dependent on mature microRNA using Dicer mutant (DICER$^{Ex5}$) HCT116 cells. It replicates Figures 3G,H, and Supplementary Figure 3B.

### Sampling

- The experiment will be repeated four times (Western blot) and three times (qRT-PCR) for a minimum power of 80%.
  - See Power Calculations section for details.
- Each experiment consists of HCT116 WT and HCT116 DICER$^{Ex5}$ cells transfected with siRNA against PTEN ceRNAs:
  - Cohort 1: siRNA against nontargeting control 2 (siNC)
  - Cohort 2: siGENOME siRNA against *SERINC1* (siSER)
  - Cohort 3: siGENOME siRNA against *VAPA* (siVAPA)
  - Cohort 4: siGENOME siRNA against *CNOT6L* (siCNO)
  - Cohort 5: siGENOME siRNA against *PTEN* (siPTEN)
  - Cohort 6: siGLO RISC-free siRNA (siGLO)
- Effects of silencing ceRNAs will be tested with
  - Western Blot for PTEN protein (Figure 3G & 3H)
  - qRT-PCR to confirm target genes were silenced (Supplementary Figure 3B)
  - siGLO fluorescence cell counts to confirm transfection efficiency

### Materials and reagents

| Reagent | Type | Manufacturer | Catalog # | Comments |
|---|---|---|---|---|
| HCT116 WT and DICER$^{Ex5}$ cells | Cells | Horizon Discovery | HD R02-019 | |
| siGLO RISC-free siRNA siGLO | siRNA | Dharmacon | D-001600-01-05 | |
| siGenome siRNA for nontargeting control 2 | siRNA | Dharmacon | D-001210-02-05 | Catalog # communicated by original authors |
| siGenome siRNA for *SERINC1* | siRNA | Dharmacon | M-010725-00-0005 | Catalog # communicated by original authors |
| siGenome siRNA for *VAPA* | siRNA | Dharmacon | M-021382-01-0005 | Catalog # communicated by original authors |
| siGenome siRNA for *CNOT6L* | siRNA | Dharmacon | M-016411-01-0005 | Catalog # communicated by original authors |
| siGenome siRNA for *PTEN* | siRNA | Dharmacon | M-003023-02-0005 | Catalog # communicated by original authors |
| Dulbecco's Modified Eagle's Medium (DMEM) | Cell Culture Reagent | Invitrogen | 10313-039 | Catalog # communicated by original authors |

*Continued on next page*

*Continued*

| Reagent | Type | Manufacturer | Catalog # | Comments |
|---|---|---|---|---|
| Fetal Bovine Serum (FBS) | Cell Culture Reagent | Invitrogen | 10438-026 | Catalog # communicated by original authors |
| Penicillin/Streptomycin | Cell Culture Reagent | Life Technologies | 15140-163 | Communicated by original authors |
| Glutamine | Cell Culture Reagent | Life Technologies | 25030-081 | Communicated by original authors |
| Trypsin | Transfection Reagent | Life Technologies | 15400-054 | Communicated by original authors |
| Dharmafect 1 | Transfection Reagent | Thermo Fisher Scientific | T200104 | Communicated by original authors |
| TRIzol reagent | qPCR reagent | Life Technologies | 15596026 | Communicated by original authors |
| RNeasy kit | qPCR reagent | Qiagen | 74104 | Communicated by original authors |
| High Capacity cDNA Archive kit | qPCR reagent | Life Technologies | 4368814 | Communicated by original authors |
| TaqMan probe *PTEN* | qPCR probes | Life Technologies | Hs02621230_s1 | |
| TaqMan probe *CNOT6L* | qPCR probes | Life Technologies | Hs00375913_m1 | |
| TaqMan probe *VAPA* | qPCR probes | Life Technologies | Hs00427749_m1 | |
| TaqMan probe *SERINC1* | qPCR probes | Life Technologies | Hs00380375_m1 | |
| TaqMan control probe *ß-ACTIN* | qPCR probes | Life Technologies | Hs00969077_m1 | Communicated by original authors |
| TaqMan Fast Advanced Master Mix | qPCR reagent | Life Technologies | 4444964 | Communicated by original authors |
| StepOne Plus Real-Time PCR system | Equipment | Applied Biosystems | | Replaces LightCycler 480 System |
| Nanodrop 2000c Spectrometer | Equipment | Thermo Scientific | | |
| PBS | Western Reagent | Life Technologies | 14190250 | Communicated by original authors |
| Lysis Buffer | Western Reagent | RIPA lysis buffer: 50mM Tris-HCl pH 7.4, 150mM NaCl, 1% NP-40, 0.5% sodium deoxycholate, 0.1% SDS, 5mM EDTA supplemented with protease inhibitors | | |
| Protease inhibitors | Western Reagent | Roche Diagnostics | 11873580001 | Communicated by original authors |
| Bradford Assay | Western Reagent | Bio-Rad | | Catalog # communicated by original authors |
| Bis-Tris acrylamide NuPAGE gels 4–15% Mini-PROTEAN TGX Precast Protein Gels | Western Reagent | Biorad | 456–1084 | Replaces NuPage gels from Life Technologies (communicated by original authors) |
| Tris-Glycine SDS PAGE buffer (10x) | Western Reagent | National Diagostic | EC-870-4L | Replaces MOPS buffer from Invitrogen |
| Nitrocellulose membranes | Western Reagent | Thermo Fisher Scientific | 45004006 | Catalog # communicated by original authors |
| 10xTBS buffer | Western Reagent | Biorad | 170–6435 | Replaces NuPage buffer from Invitrogen |
| Methanol | Reagent | Pharmco | 339000ACSCSGL | Communicated by original authors |
| Mouse anti-HSP90 monoclonal antibody (90kDa) | Antibody | Becton Dickinson | 61041 | Catalog # communicated by original authors |
| Rabbit anti-PTEN monoclonal antibody (54kDa) | Antibody | Cell Signaling | 9559 | Catalog # communicated by original authors |
| Anti-mouse HRP-conjugated secondary antibody | Antibody | Abcam | Ab6728 | Original not specified |

*Continued on next page*

*Continued*

| Reagent | Type | Manufacturer | Catalog # | Comments |
|---------|------|--------------|-----------|----------|
| Amersham ECL Western Blotting Detection Kit | Western Blot Reagent | Amersham | RPN 2108 | Replaces ECL from Applied Biological Materials |
| X-ray Film (Hyblot CL, 8x10 inch) | Western Blot Reagent | Denville | E3018 | Original not specified |
| Spectrophotometer | Equipment | Beckman Coulter | Spectra max M2 | Replaces Beckman Model DU-800 (communicated by original authors) |

## Procedure

### Notes

- All cells will be sent for mycoplasma testing and STR profiling.
- HCT116 cells (wild-type and mutant) are maintained in DMEM with 10% FBS, [#]100 U/ml penicillin/100 µg/ml streptomycin, and [#]2 mM glutamine at 37°C/5% $CO_2$ in a humidified atmosphere.

1. Transfect HCT116 cells with siRNAs:
   a. Separate HCT116 WT and DICER [Ex5] cells into four different cultures each.
      i. These will be biological replicates.
   b. For each cell type (WT and DICER Ex5) seed cells at $1.3 \times 10^5$ cells per well in 12 well dishes
      i. Seed 11 wells per replicate: 5 transfections x 2 wells each (one for Step 2, one for Step 3) and 1 transfection (siGlo) x 1 well.
      ii. Note: During the last replicate, only seed 6 wells per cell type (5 transfection conditions for Step 2) and 1 transfection condition for siGlo RISC free siRNA transfection control.
   c. Transfect cells with 100 nM siRNA (or siGLO controls) using Dharmafect 1 according to manufacturer's instructions.
      i. Note: make up a separate transfection mixture for each replicate.
      ii. Transfect a pair of wells per replicate with each of the following:
         1. siNC
         2. siSER
         3. siVAPA
         4. siCNO
         5. siPTEN
      iii. Transfect a single well per replicate with the following:
         1. siGLO
      iv. After 24-48 hr, assess number of fluorescent cells transfected with siGLO to confirm >90% transfection efficiency.
         1. If transfection is less than 90%, record efficiency, exclude replicate and omit it from the rest of the procedure. Repeat procedure until >90% efficiency is obtained.
         2. If modification to transfection is needed, record and maintain modified steps for remaining replicates.
   d. Incubate for 72 hr at 37°C in 5% $CO_2$ in a humidified atmosphere.
      i. Replace growth medium after 4 hr.
2. Using one of each pair of wells (except during replicate 4), confirm siRNA knock down with qRT-PCR as in Protocol 1 Step 3. Measure mRNA expression for each siRNA transfection sample with its appropriate target and *ß-ACTIN*, and test each probe separately using RNA from the NC control transfection.
   - *PTEN*
   - *CNOT6L*
   - *VAPA*
   - *SERINC1*
   - *β-ACTIN* [endogenous control communicated by original author]
      a. Prepare 10 µl real-time PCR reaction in triplicate for each reaction consisting of:
         1. 5 µl TaqMan mastermix
         2. 0.5 µl TaqMan probe for the gene of interest

 3. 4.5 µl cDNA (diluted 10x)
 4. Use standard TaqMan cycling protocol:
 a. 50°C 2 min
 b. 95°C 20 s
 c. 40 cycles of 95°C 1 s, 60°C 20 s
 b. Normalize each mRNA expression to *ß-ACTIN* and then normalize each siRNA to siNC for that transcript.
3. Using the second well of each pair of wells, assess PTEN protein expression by Western Blot:
 a. Wash cells in chilled PBS
 b. Lyse cells directly in wells by incubating on ice for 20 min with RIPA lysis buffer containing protease inhibitors.
 c. Clear lysates by centrifugation at 4°C for 15 min at 12,100$xg$.
 d. Determine protein concentrations with Bradford assay following manufacturer's instructions.
 e. Separate 5 µg of total protein by SDS-PAGE on 4–15% 4-15% Mini-PROTEAN TGX pre-cast protein gels in Tris-Glycine SDS PAGE buffer.
 i. HCT116 cells express high levels of PTEN protein so 5 µg should be sufficient for detection.
 f. Transfer to nitrocellulose membranes in transfer buffer containing 10% methanol for 1 hr at 40V at room temperature.
 i. *Confirm protein transfer by Ponceau staining.
 g. Block membrane with 5% milk in #TBST for 30 min.
 h. Probe membranes specific primary antibodies:
 i. PTEN: 1:1000
 ii. HSP90: 1:1000
 i. Wash membrane 3 times in 1X TBST for 5 min each on shaker.
 j. Incubate with #anti-rabbit (with PTEN primary) or #anti-mouse (for HSP90 primary) HRP conjugated secondary antibody (1:2000) for 1 hr on shaker at room temperature.
 k. Remove membrane from secondary antibody and wash three times in 1X TBST for 5 min each.
 l. Prepare ECL solution and incubate membrane.
 m. Expose membrane to X-ray film, develop and scan. Take a range of exposures (1 s, 15 s, 60 s) for each film.
 i. Note from the original author: Care should be taken not to overload the gel or to overexpose the film. ceRNA regulation may only result in a 50% increase or decrease in protein levels, this difference may be overlooked if the signal is saturated and not within the dynamic range of the film.
 n. Normalize PTEN to HSP90 for each sample.
4. Repeat 3 additional times.

## Deliverables

- Data to be collected:
  - QC image data confirming transfection efficiency by measuring the number of fluorescent cells transfected with siGLO
  - QC data for total RNA ($A_{260/280}$ and $A_{260/230}$ absorbance ratios)
  - Raw qPCR data for each sample and a graph the mean of each target gene normalized with *β-ACTIN* and normalized relative to NC control. (Compare to Supplementary Figure 3B)
  - Full scans of each western blot with ladder (Compare to Figure 3G)
  - Raw data of band analysis and normalized bands for each sample (Compare to Figure 3H)

## Confirmatory analysis plan

- Statistical Analysis of the Replication Data:
  Note: At the time of analysis, we will test for normality and homoscedasticity of the data. If the data appears skewed, we will perform the appropriate transformation to proceed with the

proposed statistical analysis. If this is not possible, we will perform the equivalent non-parametric Wilcoxon-Mann-Whitney test.

- ▪ Western blot: Two-way ANOVA of normalized PTEN levels from HCT116 cells (wild type or DICER[Ex5] cells) transfected with siRNA for SERINC1, VAPA, CNOT6L, PTEN, or control NC followed by Bonferroni-corrected planned comparisons:
  - • siNC vs. each siRNA for each cell line (8 comparisons total).
- ▪ qRT-PCR: Bonferroni corrected one-sample $t$-tests of normalized mRNA expression in HCT116 cells (wild type or DICER[Ex5] cells) transfected with siRNA against *SERINC1, VAPA, CNOT6L*, or *PTEN* compared to a constant (siNC=1) (8 comparisons total).
- • Meta-analysis of original and replication attempt effect sizes:
  - ▪ This replication attempt will perform the statistical analysis listed above, compute the effects sizes, compare them against the reported effect size in the original paper and use a meta-analytic approach to combine the original and replication effects, which will be presented as a forest plot.

## Known differences from the original study

All known differences are listed in the materials and reagents section above with the originally used item listed in the comments section. All differences have the same capabilities as the original and are not expected to alter the experimental design.

## Provisions for quality control

Extracted RNA integrity will be reported with $A_{260/280}$ and $A_{260/230}$ absorbance ratios, and transfection efficiency will be checked using the siGLO control. The cells will be sent for mycoplasma testing confirming lack of contamination and STR profiling confirming cell line authenticity. Transfection efficiency will be recorded for each replicate and any transfection that does not contain >90% efficiency will be excluded and not continue through the rest of the procedure. If the efficiency does not reach >90%, then any modifications to the transfection protocol will be recorded. qRT-PCR will be performed to confirm silencing of mRNA expression. Images of Ponceau staining to confirm protein transfer. All data obtained from the experiment - raw data, data analysis, control data, and quality control data - will be made publicly available, either in the published manuscript or as an open access dataset available on the Open Science Framework (https://osf.io/oblj1/).

## Protocol 4: Effect of knock-down of ceRNA network genes on cell proliferation

This experiment tests the effects of siRNA-mediated depletion of *PTEN, CNOT6L*, and *VAPA* expression on cell proliferation in DU145, HCT116 WT, and HCT116 DICER[Ex5] cells. It replicates Figure 5B.

### Sampling

- • This experiment will be repeated five (DU145 cells) times and four (HCT116 cells) times for a minimum power of 80%.
  - ▪ See Power Calculations section for details.
- • Each experiment consists of DU145, HCT116 WT, and HCT116 DICER[Ex5] cells transfected with siRNA against PTEN ceRNAs:
  - ▪ Cohort 1: siGLO RISC-free siRNA (siGLO)
  - ▪ Cohort 2: siRNA against nontargeting control 2 (siNC)
  - ▪ Cohort 3: siGENOME siRNA against *VAPA* (siVAPA)
  - ▪ Cohort 4: siGENOME siRNA against *CNOT6L* (siCNO)
  - ▪ Cohort 5: siGENOME siRNA against *PTEN* (siPTEN)
- • Effects of silencing ceRNAs will be tested with
  - ▪ qRT-PCR to confirm target genes were silenced [additional QC]
  - ▪ siGLO fluorescence cell counts to confirm transfection efficiency
  - ▪ Assessment of cell proliferation (Figure 5B)

## Materials and reagents

| Reagent | Type | Manufacturer | Catalog # | Comments |
|---|---|---|---|---|
| DU145 human prostate cancer cells | Cells | ATCC | HTB-81 | |
| HCT116 WT and DICER[Ex5] cells | Cells | Horizon Discovery | HD R02-019 | |
| siGLO RISC-free siRNA siGLO | siRNA | Dharmacon | D-001600-01-05 | |
| siGenome siRNA for nontargeting control 2 | siRNA | Dharmacon | D-001210-02-05 | Catalog # communicated by original authors |
| siGenome siRNA for *VAPA* | siRNA | Dharmacon | M-021382-01-0005 | Catalog # communicated by original authors |
| siGenome siRNA for *CNOT6L* | siRNA | Dharmacon | M-016411-01-0005 | Catalog # communicated by original authors |
| siGenome siRNA for *PTEN* | siRNA | Dharmacon | M-003023-02-0005 | Catalog # communicated by original authors |
| Dulbecco's Modified Eagle's Medium (DMEM) | Cell Culture Reagent | Invitrogen | 10313-039 | Catalog # communicated by original authors |
| Fetal Bovine Serum (FBS) | Cell Culture Reagent | Invitrogen | 10438-026 | Catalog # communicated by original authors |
| Penicillin/Streptomycin | Cell Culture Reagent | Life Technologies | 15140-163 | Communicated by original authors |
| Glutamine | Cell Culture Reagent | Life Technologies | 25030-081 | Communicated by original authors |
| Trypsin | Transfection Reagent | Life Technologies | 15400-054 | Communicated by original authors |
| Dharmafect 1 | Transfection Reagent | Thermo Fisher Scientific | T200104 | Communicated by original authors |
| TRIzol reagent | qPCR reagent | Life Technologies | 15596026 | Communicated by original authors |
| RNeasy kit | qPCR reagent | Qiagen | 74104 | Communicated by original authors |
| High Capacity cDNA Archive kit | qPCR reagent | Life Technologies | 4368814 | Communicated by original authors |
| TaqMan probe *PTEN* | qPCR probes | Life Technologies | Hs02621230_s1 | |
| TaqMan probe *CNOT6L* | qPCR probes | Life Technologies | Hs00375913_m1 | |
| TaqMan probe *VAPA* | qPCR probes | Life Technologies | Hs00427749_m1 | |
| TaqMan control probe *ß-ACTIN* | qPCR probes | Life Technologies | Hs00969077_m1 | Additional control |
| TaqMan Fast Advanced Master Mix | qPCR reagent | Life Technologies | 4444964 | Communicated by original authors |
| StepOne Plus Real-Time PCR system | Equipment | Applied Biosystems | | Replaces LightCycler 480 System |
| Nanodrop 2000C Spectrometer | Equipment | Thermo Scientific | | |
| PBS | Western Reagent | Life Technologies | 14190250 | Communicated by original authors |
| Formalin | Fixative | Sigma Aldrich | HT501128-4l | Communicated by original authors |
| Crystal Violet | Stain | Sigma Aldrich | C-3886 | Communicated by original authors |
| 10% acetic acid | Solubilization reagent | Thermo Fisher Scientific | A38212 | Communicated by original authors |
| BioTek Synergy HT Multi-mode Microplate Reader | Equipment | BioTek Instrument | | Replaces Beckman Coulter Model DU-800 (communicated by original authors) |

## Procedure

Notes:

- HCT116 cells (wild-type and mutant) are maintained in DMEM with 10% FBS, #100 U/ml penicillin/100 µg/ml streptomycin, and #2 mM glutamine at 37°C/5% $CO_2$ in a humidified atmosphere.
- DU145 cells are maintained in DMEM supplemented with 10% FBS, #100 U/ml penicillin/100 µg/ml streptomycin, and #2 mM glutamine at 37°C in 5% $CO_2$ in a humidified atmosphere.
- All cells will be sent for mycoplasma testing and STR profiling.

1. Transfect DU145, HCT116 WT, and HCT116 DICER[Ex5] cells with siRNAs
   a. Separate DU145 into five cultures each, and HCT116 WT, and HCT116 DICER[Ex5] cells each into 4 different cultures.
      i. These will be biological replicates for each cell line.
   b. Seed cells 1.3 x $10^5$ cells per well of a 12-well plate for subsequent experiments:
      i. For measuring transfection efficiency (Step 1c ii):
         1. Seed 1 well (Cohort 1) per replicate per cell line.
            a. 5 wells for DU145 cells
            b. 4 wells for HCT116 WT cells
            c. 4 wells for HCT116 Dicer[Ex5] cells
      ii. For cell proliferation assay (Step 2).
         1. Seed 4 wells (Cohorts 2-5) per replicate per cell line.
            a. 20 wells for DU145 cells
            b. 16 wells for HCT116 WT cells
            c. 16 wells for HCT116 Dicer[Ex5] cells
      iii. For qPCR confirmation of siRNA knockdown (Step 3).
         1. Seed 4 wells (Cohorts 2-5) per replicate per cell line.
            a. 20 wells for DU145 cells
            b. 16 wells for HCT116 WT cells
            c. 16 wells for HCT116 Dicer[Ex5] cells
   c. Transfect wells with 100 nM of appropriate siRNA using Dharmafect1 according to manufacturer's instructions.
      i. Note: make up a separate transfection mix for each biological replicate.
      ii. Incubate wells for measuring transfection efficiency for 24-48 hr, then assess number of fluorescent cells transfected with siGLO to confirm >90% transfection efficiency.
         1. If transfection is less than 90%, record efficiency for attempt, exclude attempt and do not continue with the rest of the procedure. Repeat procedure until >90% efficiency is obtained.
         2. If modification to transfection is needed during first attempt(s), record and maintain modified steps for remaining replicates.
      iii. Incubate wells for seeding the cell proliferation assay for 8 hr, then proceed to Step 2.
      iv. Incubate wells for qPCR for 72 hr, then proceed to Step 3.
2. Measure cell proliferation
   a. Eight hours after transfection, trypsinize and resuspend cells. Split each well into 1 well each of four 12-well plates, seeding 20,000 cells/well. Incubate overnight.
      i. Two 12-well plates (a set) will provide sufficient wells to accommodate all replicates for one day of the time course per cell line.
      ii. 8 plates will be needed per cell line for a full 4 day time course.
   b. Starting on the following day (d0), fix one set of plates per cell line per day.
      i. Wash cells with PBS.
      ii. Fix cells in 10% formalin solution for 10 min at room temperature.
      iii. Store cells in PBS at 4°C until all plates are fixed.
         1. Plates should be collected on day 0, 1, 2 and 3.
   c. c. On day 3, stain all wells of all plates with crystal violet.
      i. Add 1 ml 0.1% Crystal Violet solution in 20% methanol.
      ii. Shake gently for 15 min at room temperature.
      iii. Wash 2 times in distilled water and let plates dry completely.
      iv. Solubilize remaining crystal violet by adding 1 ml of 10% acetic acid to each well.
      v. Shake gently for 15 min at room temperature.
      vi. Transfer 100 µl to a 96-well plate and measure OD at 595 nm in a plate reader.
3. Confirm siRNA knock down with qPCR as in Protocol 1 Step 3. Perform qRT-PCR to measure mRNA expression for each siRNA transfection sample with its appropriate target and *ß-ACTIN*, and test each probe separately using RNA from the NC control transfection.

- *PTEN*
- *CNOT6L*
- *VAPA*
- *ß-ACTIN* [endogenous control communicated by original author]
  1. Prepare 10 µl real-time PCR reaction in triplicate for each reaction consisting of:
     a. 5 µl TaqMan mastermix
     b. 0.5 µl TaqMan probe for the gene of interest
     c. 4.5 µl cDNA (diluted 10x)
     d. Use standard TaqMan cycling protocol:
        i. 50°C 2 min
        ii. 95°C 20 s
        iii. 40 cycles of 95°C 1 s, 60°C 20 s

## Deliverables

- Data to be collected:
  - QC image data confirming transfection efficiency by measuring the number of fluorescent cells transfected with siGLO
  - QC data for total RNA ($A_{260/280}$ and $A_{260/230}$ absorbance ratios)
  - Raw qPCR data for each sample and a graph of the mean of each target gene normalized with *ß-ACTIN* and graphed relative to NC control.
  - Raw numbers for optical density measures of colonies for each sample.

## Confirmatory analysis plan

- Statistical Analysis of the Replication Data:
  Note: At the time of analysis, we will test for normality and homoscedasticity of the data. If the data appears skewed, we will perform the appropriate transformation in order to proceed with the proposed statistical analysis. If this is not possible we will perform the equivalent non-parametric Wilcoxon-Mann-Whitney test.
  - Cell proliferation data: One-way ANOVA of AUC values of DU145 cells transfected with siRNA for *VAPA, CNOT6L, PTEN*, or siNC followed by Bonferroni-corrected planned comparisons:
    - siNC vs each siRNA (3 comparisons total).
  - Cell proliferation data: Two-way ANOVA of AUC values of HCT116[WT] or HCT116 DICER[Ex5] cells transfected with siRNA for *VAPA, CNOT6L, PTEN*, or siNC followed by Bonferroni-corrected planned comparisons:
    - siNC vs. each siRNA, for each cell line (6 comparisons total).
- Meta-analysis of original and replication attempt effect sizes:
  - This replication attempt will perform the statistical analysis listed above, compute the effects sizes, compare them against the reported effect size in the original paper and use a meta-analytic approach to combine the original and replication effects, which will be presented as a forest plot.
- Additional exploratory analysis:
  siRNA knockdown confirmation [additional control]
  - Two-way ANOVA of mRNA expression in HCT116 cells (wild type or DICER[Ex5] cells) transfected with siRNA against NC, *VAPA, CNOT6L*, or *PTEN*, with the following Bonferroni-corrected comparisons:
    - Non-coding siRNA vs. each of the ceRNA transfected cells (3 comparisons total).

## Known differences from the original study

All known differences are listed in the materials and reagents section above, with the originally used item listed in the comments section. All differences have the same capabilities as the original and are not expected to alter the experimental design.

## Provisions for quality control

Extracted RNA integrity will be reported with $A_{260/280}$ and $A_{260/230}$ absorbance ratios, and transfection efficiency will be checked using the siGLO control. Cells will be sent for mycoplasma testing confirming lack of contamination and STR profiling confirming cell line authenticity. Transfection

efficiency will be recorded for each replicate and any transfection that does not contain >90% efficiency will be excluded and not continue through the rest of the procedure. Any modifications to the transfection protocol will be recorded and the procedure will be maintained for the remaining replicates. All data obtained from the experiment - raw data, data analysis, control data and quality control data - will be made publicly available, either in the published manuscript or as an open access dataset available on the Open Science Framework (https://osf.io/oblj1/).

## Protocol 5: Knock-down of ceRNA network genes results in AKT activation

This experiment tests the effects of siRNA-mediated depletion of *PTEN, CNOT6L*, and *VAPA* expression on AKT activation in DU145, HCT116 WT, and HCT116 Dicer[Ex5] cells. It replicates Figure 5A.

### Sampling

- This experiment will be repeated at least 7 times for a minimum power of 80%. The original Western blot data is qualitative, thus to determine an appropriate number of replicates to initially perform, sample sizes based on a range of potential variance was determined.
    - See Power Calculations section for details.
- Each experiment consists of DU145, HCT116 WT, and HCT116 DICER[Ex5] cells transfected with siRNA against *PTEN* ceRNAs:
    - Cohort 1: siGLO RISC-free siRNA (siGLO)
    - Cohort 2: siRNA against nontargeting control 2 (siNC)
    - Cohort 3: siGENOME siRNA against *VAPA* (siVAPA)
    - Cohort 4: siGENOME siRNA against *CNOT6L* (siCNO)
    - Cohort 5: siGENOME siRNA against *PTEN* (siPTEN)
- Effects of silencing ceRNAs will be tested with
    - qRT-PCR to confirm target genes were silenced [additional QC]
    - siGLO fluorescence cell counts to confirm transfection efficiency
    - Assessment of AKT phosphorylation by Western blot (Figure 5A)

### Materials and reagents

| Reagent | Type | Manufacturer | Catalog # | Comments |
|---|---|---|---|---|
| DU145 human prostate cancer cells | Cells | ATCC | HTB-81 | |
| HCT116 WT and DICER[Ex5] cells | Cells | Horizon Discovery | HD R02-019 | |
| siGLO RISC-free siRNA siGLO | siRNA | Dharmacon | D-001600-01-05 | |
| siGenome siRNA for nontargeting control 2 | siRNA | Dharmacon | D-001210-02-05 | Catalog # communicated by original authors |
| siGenome siRNA for *VAPA* | siRNA | Dharmacon | M-021382-01-0005 | Catalog # communicated by original authors |
| siGenome siRNA for *CNOT6L* | siRNA | Dharmacon | M-016411-01-0005 | Catalog # communicated by original authors |
| siGenome siRNA for *PTEN* | siRNA | Dharmacon | M-003023-02-0005 | Catalog # communicated by original authors |
| TaqMan probe *PTEN* | qPCR probes | Life Technologies | Hs02621230_s1 | |
| TaqMan probe *CNOT6L* | qPCR probes | Life Technologies | Hs00375913_m1 | |
| TaqMan probe *VAPA* | qPCR probes | Life Technologies | Hs00427749_m1 | |
| TaqMan control probe *ß-ACTIN* | qPCR probes | Life Technologies | Hs00969077_m1 | Additional control |
| TaqMan Fast Advanced Master Mix | qPCR reagent | Life Technologies | 4444964 | Communicated by original authors |
| StepOne Plus Real-Time PCR system | Equipment | Applied Biosystems | | Replaces LightCycler 480 System |
| Nanodrop 2000C Spectrometer | Equipment | Thermo Scientific | | |

*Continued on next page*

*Continued*

| Reagent | Type | Manufacturer | Catalog # | Comments |
|---|---|---|---|---|
| Dulbecco's Modified Eagle's Medium (DMEM) | Cell Culture Reagent | Invitrogen | 10313-039 | Catalog # communicated by original authors |
| Fetal Bovine Serum (FBS) | Cell Culture Reagent | Invitrogen | 10438-026 | Catalog # communicated by original authors |
| Penicillin/Streptomycin | Cell Culture Reagent | Life Technologies | 15140-163 | Communicated by original authors |
| Glutamine | Cell Culture Reagent | Life Technologies | 25030-081 | Communicated by original authors |
| Trypsin | Transfection Reagent | Life Technologies | 15400-054 | Communicated by original authors |
| Dharmafect 1 | Transfection Reagent | Thermo Fisher Scientific | T200104 | Communicated by original authors |
| PBS | Western Reagent | Life Technologies | 14190250 | Communicated by original authors |
| Lysis Buffer | Western Reagent | RIPA lysis buffer: 50mM Tris-HCl pH 7.4, 150mM NaCl, 1% NP-40, 0.5% sodium deoxycholate, 0.1% SDS, 5mM EDTA supplemented with proteinase inhibitors | | |
| Protease inhibitors | Western Reagent | Roche Diagnostics | 11873580001 | Communicated by original authors |
| Bradford Dye | Western Reagent | Bio-Rad | 500-0006 | Catalog # communicated by original authors |
| 4–15% Mini-PROTEAN TGX Precast Protein Gels | Western Reagent | Biorad | 456–1084 | Replaces NuPage gels from Life Technologies (communicated by original authors) |
| Tris-Glycine SDS PAGE buffer (10x) | Western Reagent | National Diagnostic | EC-870-4L | Replaces MOPS buffer from Invitrogen |
| Nitrocellulose membranes | Western Reagent | Thermo Fisher Scientific | 45004006 | Catalog # communicated by original authors |
| 10xTBS buffer | Western Reagent | Biorad | 170–6435 | Replaces NuPage buffer from Invitrogen |
| Methanol | Chemical | Pharmco | 339000ACSCSGL | Communicated by original authors |
| Rabbit anti-pAKT (Ser473) polyclonal antibody (60kDa) | Antibody | Cell Signaling | 9271 | Catalog # communicated by original authors |
| Rabbit anti-AKT polyclonal antibody (60kDa) | Antibody | Cell Signaling | 9272 | Catalog # communicated by original authors |
| Amersham ECL Western Blotting Detection Kit | Western Blot Reagent | Amersham | RPN2108 | Replaces ECL from Applied Biological Materials |
| Spectrophotometer | Equipment | Beckman Coulter | Spectra max M2 | Replaces Beckman Model DU-800 (communicated by original authors) |

## Procedure

Notes:

- HCT116 cells (wild-type and mutant) are maintained in DMEM with 10% FBS, [#]100 U/ml penicillin/100 µg/ml streptomycin, and [#]2 mM glutamine at 37°C/5% $CO_2$ in a humidified atmosphere.
- DU145 cells are maintained in DMEM supplemented with 10% FBS, [#]100 U/ml penicillin/100 µg/ml streptomycin, and [#]2 mM glutamine at 37°C in 5% $CO_2$ in a humidified atmosphere.
- All cells will be sent for mycoplasma testing and STR profiling.

1. Transfect DU145, HCT116 WT, and HCT116 DICER[Ex5] cells with siRNAs
   a. Seed cells for subsequent experiments with $1.3 \times 10^5$ cells per well in a 12-well plate:
      i. For measuring transfection efficiency (Step 1c ii):

1. Seed 1 well (Cohort 1) per replicate.
   a. DU145 cells
   b. HCT116 WT cells
   c. HCT116 Dicer^Ex5 cells

   ii. For AKT activation and Western blot (Step 2).
   1. Seed 3 well (Cohort 2-5) per replicate.
      a. 12 wells for DU145 cells
      b. 12 wells for HCT116 WT cells
      c. 12 wells for HCT116 Dicer^Ex5 cells

   b. Transfect wells with 100 nM of appropriate siRNA using Dharmafect1 according to manufacturer's instructions.
   i. Note: make up a separate transfection mix for each biological replicate.
   ii. Incubate wells for measuring transfection efficiency for 24-48 hr, then assess number of fluorescent cells transfected with siGLO to confirm >90% transfection efficiency.
      1. If transfection is less than 90%, record efficiency, exclude attempt and do not continue with the rest of the procedure. Repeat procedure until >90% efficiency is obtained.
      2. If modification to transfection is needed, record and maintain modified steps for remaining replicates.

2. Stimulate activation of AKT then measure levels of phosphorylated AKT by Western blot.
   a. After 72 hr, serum-starve cells overnight: replace media with serum-free media and incubate overnight (approximately 16 hr).
   b. The following morning, harvest one well at 0 min (pre-stimulation), re-stimulate the remaining cells by adding the appropriate volume of warmed 100% FBS to existing media in each trio of matched wells for a 10% final concentration. Incubate wells for 5 or 15 min.
   i. Harvest one well at 5 min and one well at 15 min post FBS addition.
   c. Harvest cells and perform Western blot as specified in Protocol 3 step 3.
   i. Note: load 10 μg of protein per well.
   ii. Probe membranes specific primary antibodies
      1. pAKT (Ser473); 1:1000
      2. total AKT; 1:1000
         a. Loading control
   iii. Note from original author: Phosphorylated proteins are less stable in lysis buffer than non-phosphorylated proteins. Try to use fresh lysates for subsequent western blotting as far as possible. Transfer samples to the protein loading buffer as fast as possible and keep freeze thaw cycles to an absolute minimum.
   d. Normalize pAKT to total AKT for each sample.

3. Repeat at least 6 additional times.

## Deliverables

- Data to be collected:
  - QC image data confirming transfection efficiency by measuring the number of fluorescent cells transfected with siGLO
  - QC data for total RNA ($A_{260/280}$ and $A_{260/230}$ absorbance ratios)
  - Raw qPCR data for each sample and a graph of the mean of each target gene normalized with ß-ACTIN and graphed relative to NC control.
  - Full scans of all films for each western including ladder.

## Confirmatory analysis plan

- Statistical Analysis of the Replication Data:
  Note: At the time of analysis, we will test for normality and homoscedasticity of the data. If the data appears skewed, we will perform the appropriate transformation to proceed with the proposed statistical analysis. If this is not possible, we will perform the equivalent non-parametric Wilcoxon-Mann-Whitney test.
  - Two-way ANOVA of normalized pAKT levels of DU145 cells transfected with siRNA for *VAPA, CNOT6L, PTEN*, or siNC measured at 0 min, 5 min, and 15 min followed by Bonferroni-corrected planned contrasts:
    - siNC vs each siRNA, collapsed across all times (3 contrasts total).

- Three-way ANOVA (3x4x2) of normalized pAKT levels of HCT116^WT or HCT116 DICER^Ex5 cells transfected with siRNA for *VAPA, CNOT6L, PTEN*, or siNC measured at 0 min, 5 min, and 15 min:
  - HCT116^WT cells with the following Bonferroni-corrected planned contrasts:
    - siNC vs. each siRNA, collapsed across all times (3 contrasts total).
  - HCT116 DICER^Ex5 cells with the following Bonferroni-corrected planned contrasts:
    - siNC vs. each siRNA, collapsed across all times (3 contrasts total).
- Meta-analysis of original and replication attempt effect sizes:
  - The replication data (mean and 95% confidence interval) will be plotted with the original reported data value plotted as a single point on the same plot for comparison.

## Known differences from the original study

All known differences are listed in the materials and reagents section above, with the originally used item listed in the comments section. All differences have the same capabilities as the original and are not expected to alter the experimental design.

## Provisions for quality control

The cells will be sent for mycoplasma testing confirming lack of contamination and STR profiling confirming cell line authenticity. Transfection efficiency will be recorded for each replicate and any transfection that does not contain >90% efficiency will be excluded and not continue through the rest of the procedure. Any modifications to the transfection protocol will be recorded, and the procedure will be maintained for the remaining replicates. Images of Ponceau staining to confirm protein transfer. All data obtained from the experiment - raw data, data analysis, control data, and quality control data - will be made publicly available, either in the published manuscript or as an open access dataset available on the Open Science Framework (https://osf.io/oblj1/).

## Power calculations

For additional details on power calculations, please see analysis scripts and associated files on the Open Science Framework:

https://osf.io/c8hb5

## Protocol 1

Summary of original luciferase activity data:

- Note: data provided by original authors for Figure 3C

| siRNA | Luciferase activity | SD | N |
|-------|--------------------|------|---|
| siNC | 100 | 9.28 | 4 |
| siSER | 70.29 | 6.99 | 4 |
| siZNF | 108.62 | 9.2 | 4 |
| siVAPA | 47.54 | 2.89 | 4 |
| siCNO | 69.82 | 3.69 | 4 |
| siPTEN | 20.32 | 1.11 | 4 |

## Test family

- 2 tailed *t* test, Wilcoxon-Mann-Whitney test, Bonferroni's correction: alpha error = 0.01

Power Calculations performed with G*Power software, version 3.1.7 (*Faul et al., 2007*).

| Group 1 | Group 2 | Effect size $d$ | A priori power | Group 1 sample size | Group 2 sample size |
|---------|---------|-----------------|----------------|---------------------|---------------------|
| siNC | siSER | 3.61647 | 81.9% | 4 | 4 |
| siNC | siZNF | 0.9329 | 80.2%[2] | 30[2] | 30[2] |
| siNC | siVAPA | 7.63300 | 98.5%[1] | 3[1] | 3[1] |
| siNC | siCNO | 4.27377 | 93.2% | 4 | 4 |
| siNC | siPTEN | 12.0568 | 99.9%[1] | 3[1] | 3[1] |

[1] 4 samples per group will be used making the power >99.9%.

[2] A sensitivity calculation was performed since the original data showed a non-significant effect. The effect size that can be detected with 80% power and a sample size n=4 per group is 3.5378.

## Test family

- Due to the large variance, the following parametric tests are only used for comparison purposes. The sample size is based on the non-parametric tests listed above.
- ANOVA: Fixed effects, omnibus, one-way: alpha error = 0.05

## Power calculations

- Performed with G*Power software, version 3.1.7 (*Faul et al., 2007*).
- ANOVA F test statistic and partial $\eta^2$ performed with R software, version 3.1.2 (*R Core Team 2015*).

| Groups | F test statistic | Partial $\eta^2$ | Effect size $f$ | A priori power | Total sample size |
|--------|------------------|------------------|-----------------|----------------|-------------------|
| siRNA silencing groups | $F_{(5,18)=}106.0$ | 0.9672 | 5.4302 | >99.9% | 12[1] |

[1] 24 total samples (4 per group) will be used based on the planned comparisons making the power >99.9%.

## Test family

- Two-tailed $t$ test, difference between two independent means, Bonferroni correction: alpha error = 0.01

## Power calculations

- Performed with G*Power software, version 3.1.7 (*Faul et al., 2007*).

| Group 1 | Group 2 | Effect size $d$ | A priori power | Group 1 sample size | Group 2 sample size |
|---------|---------|-----------------|----------------|---------------------|---------------------|
| siNC | siSER | 3.61647 | 85.8% | 4 | 4 |
| siNC | siZNF | 0.9329 | 80.8%[2] | 29[2] | 29[2] |
| siNC | siVAPA | 7.63300 | 99.4%[1] | 3[1] | 3[1] |
| siNC | siCNO | 4.27377 | 95.4% | 4 | 4 |
| siNC | siPTEN | 12.0568 | 99.9%[1] | 3[1] | 3[1] |

[1] 4 samples per group will be used making the power >99.9%.

[2] A sensitivity calculation was performed since the original data showed a non-significant effect. The effect size that can be detected with 80% power and a sample size n=4 per group is 3.3711.

Summary of original qPCR gene expression data:

- Note: data provided by original authors for Figure S3A
- We estimated SD to be 0.001, when it was reported as zero.

| siRNA | mRNA expression | SD | Assumed N |
|---|---|---|---|
| siSER | 0.03 | 0.001 | 4 |
| siZNF | 0.35 | 0.11 | 4 |
| siVAPA | 0.03 | 0.001 | 4 |
| siCNO | 0.07 | 0.01 | 4 |
| siPTEN | 0.1 | 0.04 | 4 |

## Test family

- 2 tailed *t* test, Wilcoxon-Signed Ranks one-sample test, Bonferroni's correction: alpha error = 0.01

Power Calculations performed with G*Power software, version 3.1.7 (*Faul et al., 2007*).

| Group | Effect size *d* | A priori power | Sample size |
|---|---|---|---|
| siSER | 970.00 | 99.9% | 3 |
| siZNF | 5.91 | 97.7% | 4 |
| siVAPA | 970.00 | 99.9% | 3 |
| siCNO | 93.00 | 99.9% | 3 |
| siPTEN | 22.50 | 99.9% | 3 |

## Test family

- Due to the large variance, the following parametric tests are only used for comparison purposes. The sample size is based on the non-parametric tests listed above.
- Two-tailed *t* test, difference from a constant, Bonferroni correction: alpha error = 0.01

## Power calculations

- Performed with G*Power software, version 3.1.7 (*Faul et al., 2007*).

| Group | Effect size *d* | A priori power | sample size |
|---|---|---|---|
| siSER | 970.00 | 99.9% | 3 |
| siZNF | 5.91 | 99.0% | 4 |
| siVAPA | 970.00 | 99.9% | 3 |
| siCNO | 93.00 | 99.9% | 3 |
| siPTEN | 22.50 | 99.9% | 3 |

## Protocol 2

Summary of original Luciferase data:

- Note: data provided by original authors for Figure 3D.

| siRNA | Luciferase Activity | SD | N |
|---|---|---|---|
| Empty Vector | 100 | 8.83 | 4 |
| SER 3'U | 127.86 | 11.59 | 4 |
| VAPA 3'U1 | 140.84 | 17.8 | 4 |
| VAPA 3'U2 | 150.25 | 9.37 | 4 |
| CNO 3'U1 | 142.91 | 9.92 | 4 |
| CNO 3'U2 | 145.88 | 10.59 | 4 |
| PTEN 3'U | 153.32 | 2.06 | 4 |

## Test family

- 2 tailed *t* test, Wilcoxon-Mann-Whitney test, Bonferroni's correction: alpha error = 0.00833

Power Calculations performed with G*Power software, version 3.1.7 (*Faul et al., 2007*).

| Group 1 | Group 2 | Effect size *d* | A priori power | Group 1 sample size | Group 2 sample size |
|---|---|---|---|---|---|
| Empty Vector | SER 3'U | 2.70411 | 85.7% | 6 | 6 |
| Empty Vector | VAPA 3'U1 | 2.90675 | 91.0% | 6 | 6 |
| Empty Vector | VAPA 3'U2 | 5.51955 | 81.3%[1] | 3[1] | 3[1] |
| Empty Vector | CNO 3'U1 | 4.56935 | 94.6%[1] | 4[1] | 4[1] |
| Empty Vector | CNO 3'U2 | 4.70574 | 95.7%[1] | 4[1] | 4[1] |
| Empty Vector | PTEN 3'U | 8.31642 | 99.0%[1] | 3[1] | 3[1] |

[1] 6 samples per group will be used making the power >99.9%.

## Test family

- Due to the large variance, the following parametric tests are only used for comparison purposes. The sample size is based on the non-parametric tests listed above.
- ANOVA: Fixed effects, omnibus, one-way: alpha error = 0.05

## Power calculations

- Performed with G*Power software, version 3.1.7 (*Faul et al., 2007*).
- ANOVA F test statistic and partial $\eta^2$ performed with R software, version 3.1.2 (*R Core Team 2015*).

| Groups | F test statistic | Partial $\eta^2$ | Effect size *f* | A priori power | Total sample size |
|---|---|---|---|---|---|
| PTEN ceRNAs 3'UTRs | $F_{(6, 21)}=11.347$ | 0.7643 | 5.4302 | 99.9% | 14[1] |

[1] 42 total samples (6 per group) will be used based on the planned comparisons making the power >99.9%.

## Test family

- Two-tailed *t* test, difference between two independent means, Bonferroni correction: alpha error = 0.00833

## Power calculations

- Performed with G*Power software, version 3.1.7 (*Faul et al., 2007*).

| Group 1 | Group 2 | Effect size d | A priori power | Group 1 sample size | Group 2 sample size |
|---------|---------|---------------|----------------|---------------------|---------------------|
| Empty Vector | SER 3'U | 2.70411 | 88.5% | 6 | 6 |
| Empty Vector | VAPA 3'U1 | 2.90675 | 82.4%[1] | 5[1] | 5[1] |
| Empty Vector | VAPA 3'U2 | 5.51955 | 86.9%[2] | 3[2] | 3[2] |
| Empty Vector | CNO 3'U1 | 4.56935 | 96.5%[2] | 4[2] | 4[2] |
| Empty Vector | CNO 3'U2 | 4.70574 | 97.4%[2] | 4[2] | 4[2] |
| Empty Vector | PTEN 3'U | 8.31642 | 99.6%[2] | 3[2] | 3[2] |

[1] 6 samples per group will be used making the power 96.2%.
[2] 6 samples per group will be used making the power 99.9%.

## Protocol 3

Summary of original Western blot data:

- Note: data provided by original authors for Figure 3H.

| siRNA | Cell type | PTEN expression | SD | N |
|-------|-----------|-----------------|-----|---|
| siNC | WT | 100 | 8.3 | 4 |
| | DicerEx5 | 100 | 4.8 | 4 |
| siSER | WT | 52.6 | 8.9 | 4 |
| | DicerEx5 | 117 | 6.5 | 4 |
| SiVAPA | WT | 51.7 | 6.5 | 4 |
| | DicerEx5 | 107.5 | 9.4 | 4 |
| siCNO | WT | 58.7 | 4.5 | 4 |
| | DicerEx5 | 113 | 4.4 | 4 |
| siPTEN | WT | 1.9 | 0.2 | 4 |
| | DicerEx5 | 1.3 | 0.001 | 4 |

## Test family

- 2 tailed $t$ test, Wilcoxon-Mann-Whitney test, Bonferroni's correction: alpha error = 0.00625

Power Calculations performed with G*Power software, version 3.1.7 (*Faul et al., 2007*).

| Group 1 | Group 2 | Effect size d | A priori power | Group 1 sample size | Group 2 sample size |
|---------|---------|---------------|----------------|---------------------|---------------------|
| WT siNC | WT siSER | 5.50828 | 98.4% | 4 | 4 |
| WT siNC | WT siVAPA | 6.47928 | 87.2%[1] | 3[1] | 3[1] |
| WT siNC | WT siCNO | 6.18627 | 83.9%[1] | 3[1] | 3[1] |
| WT siNC | WT siPTEN | 16.71013 | 99.9%[1] | 3[1] | 3[1] |
| Sensitivity Calculations | | Detectable Effect size d | A priori power | Group 1 sample size | Group 2 sample size |
| Dicer siNC | Dicer siSER | 3.895 | 80% | 4 | 4 |
| Dicer siNC | Dicer siVAPA | 3.895 | 80% | 4 | 4 |
| Dicer siNC | Dicer siCNO | 3.895 | 80% | 4 | 4 |
| Dicer siNC | Dicer siPTEN | 3.895 | 80% | 4 | 4 |

[1] 4 samples per group will be used making the power >99%.

## Test family

- Due to the large variance, the following parametric tests are only used for comparison purposes. The sample size is based on the non-parametric tests listed above.
- Two-way ANOVA: Fixed effects, main effects, special and interactions: alpha error = 0.05

## Power calculations

- Performed with G*Power software, version 3.1.7 (*Faul et al., 2007*).
- ANOVA F test statistic and partial $\eta^2$ performed with R software, version 3.1.2 (*Team 2014*; *R Core Team 2015*).

| Groups | F test statistic | Partial $\eta^2$ | Effect size $f$ | A priori power | Total sample size |
|---|---|---|---|---|---|
| siRNA silencing groups in WT or DicerEx5 cells | $F_{(4,30)}$= 54.237 (interaction) | 0.87852 | 2.6892 | 89.9%[1] | 14[1] |

[1] 40 total samples (4 per group) will be used based on the planned comparisons making the power >99.9%.

## Test family

- Two-tailed *t* test, difference between two independent means, Bonferroni correction: alpha error = 0.00625

## Power calculations

- Performed with G*Power software, version 3.1.7 (*Faul et al., 2007*).

| Group 1 | Group 2 | Effect size $d$ | A priori power | Group 1 sample size | Group 2 sample size |
|---|---|---|---|---|---|
| WT siNC | WT siSER | 5.50828 | 81.1% | 3[1] | 3[1] |
| WT siNC | WT siVAPA | 6.47928 | 92.0% | 3[1] | 3[1] |
| WT siNC | WT siCNO | 6.18627 | 89.5% | 3[1] | 3[1] |
| WT siNC | WT siPTEN | 16.71013 | 82.6%[1] | 2[1] | 2[1] |
| Sensitivity Calculations | | Detectable Effect size $d$ | A priori power | Group 1 sample size | Group 2 sample size |
| Dicer siNC | Dicer siSER | 3.697 | 80% | 4 | 4 |
| Dicer siNC | Dicer siVAPA | 3.697 | 80% | 4 | 4 |
| Dicer siNC | Dicer siCNO | 3.697 | 80% | 4 | 4 |
| Dicer siNC | Dicer siPTEN | 3.697 | 80% | 4 | 4 |

[1] 4 samples per group will be used making the power >99.9%.

Summary of original mRNA expression data:

- Note: data provided by original authors for Figure S3B.

| siRNA | Cell Type | mRNA expression | SD | N |
|---|---|---|---|---|
| siSER | WT | 0.036 | 0.0049 | 4 |
| | DicerEx5 | 0.028 | 0.0007 | 4 |
| SiVAPA | WT | 0.027 | 0.0019 | 4 |
| | DicerEx5 | 0.034 | 0.0005 | 4 |

*Continued on next page*

*Continued*

| siRNA | Cell Type | mRNA expression | SD | N |
|-------|-----------|-----------------|-----|---|
| siCNO | WT | 0.107 | 0.033 | 4 |
|       | DicerEx5 | 0.033 | 0.0025 | 4 |
| siPTEN | WT | 0.075 | 0.0237 | 4 |
|        | DicerEx5 | 0.115 | 0.0414 | 4 |

## Test family

- 2 tailed *t* test, Wilcoxon-Signed Ranks one-sample test, Bonferroni's correction: alpha error = 0.00625

## Power calculations

- Performed with G*Power software, version 3.1.7 (*Faul et al., 2007*).

| Group | Effect size *d* | A priori power | Group 1 sample size |
|-------|-----------------|----------------|---------------------|
| WT siSER | 196.75 | 99.5% | 3 |
| WT siVAPA | 512.00 | 99.5% | 3 |
| WT siCNO | 27.06 | 99.5% | 3 |
| WT siPTEN | 39.03 | 99.5% | 3 |
| Dicer siSER | 1388.50 | 99.5% | 3 |
| Dicer siVAPA | 1932.00 | 99.5% | 3 |
| Dicer siCNO | 386.80 | 99.5% | 3 |
| Dicer siPTEN | 21.38 | 99.5% | 3 |

## Test family

- Due to the large variance, the following parametric tests are only used for comparison purposes. The sample size is based on the non-parametric tests listed above.
- Two-tailed *t* test, difference from a constant (mu=1), Bonferroni correction: alpha error = 0.00625

## Power calculations

- Performed with G*Power software, version 3.1.7 (*Faul et al., 2007*).

| Group | Effect size *d* | A priori power | Group 1 sample size |
|-------|-----------------|----------------|---------------------|
| WT siSER | 196.75 | 99.9% | 3 |
| WT siVAPA | 512.00 | 99.9% | 3 |
| WT siCNO | 27.06 | 99.9% | 3 |
| WT siPTEN | 39.03 | 99.9% | 3 |
| Dicer siSER | 1388.50 | 99.9% | 3 |
| Dicer siVAPA | 1932.00 | 99.9% | 3 |
| Dicer siCNO | 386.80 | 99.9% | 3 |
| Dicer siPTEN | 21.38 | 99.9% | 3 |

## Protocol 4

Summary of original cell proliferation data:

- Note: data of mean values provided by original authors for Figure 5B.

| Cell Type | siRNA | Cell Proliferation (Optical Density) | | | |
|---|---|---|---|---|---|
| | | Day 0 | Day 1 | Day 2 | Day 3 |
| DU145 | siNC | 0 | 0.08 | 0.37 | 0.92 |
| | siPTEN | 0 | 0.16 | 0.83 | 1.96 |
| | siCNO | 0 | 0.06 | 0.63 | 1.66 |
| | siVAPA | 0 | 0.12 | 0.78 | 1.75 |
| HCT116 WT | siNC | 0 | 0.30 | 0.91 | 1.35 |
| | siPTEN | 0 | 0.60 | 1.63 | 2.07 |
| | siCNO | 0 | 0.77 | 1.98 | 2.19 |
| | siVAPA | 0 | 0.66 | 1.65 | 1.98 |
| HCT116 Dicer Ex5 | siNC | 0 | 0.12 | 0.49 | 0.74 |
| | siPTEN | 0 | 0.69 | 1.72 | 1.90 |
| | siCNO | 0 | 0.49 | 1.09 | 1.75 |
| | siVAPA | 0 | 0.30 | 0.95 | 1.34 |

- Area under the curve calculation with R software, version 3.1.2 (*R Core Team 2015*).

| Cell Type | siRNA | Area under the curve | SD | N |
|---|---|---|---|---|
| DU145 | siNC | 0.910 | 0.235 | 3 |
| | siPTEN | 1.970 | 0.140 | 3 |
| | siCNO | 1.520 | 0.141 | 3 |
| | siVAPA | 1.775 | 0.076 | 3 |
| HCT116 WT | siNC | 1.885 | 0.180 | 3 |
| | siPTEN | 3.265 | 0.156 | 3 |
| | siCNO | 3.845 | 0.290 | 3 |
| | siVAPA | 3.300 | 0.275 | 3 |
| HCT116 Dicer Ex5 | siNC | 0.980 | 0.012 | 3 |
| | siPTEN | 3.360 | 0.310 | 3 |
| | siCNO | 2.455 | 0.145 | 3 |
| | siVAPA | 1.920 | 0.285 | 3 |

## DU145 cells

Test family

- 2 tailed, Wilcoxon-Mann-Whitney test, Bonferroni's correction: alpha error = 0.0167

Power Calculations performed with G*Power software, version 3.1.7 (*Faul et al., 2007*).

| Group 1 | Group 2 | Effect size *d* | A priori power | Group 1 sample size | Group 2 sample size |
|---|---|---|---|---|---|
| siNC | siPTEN | 5.115 | 89.5% | 3 | 3 |
| siNC | siCNO | 2.944 | 89.4% | 5 | 5 |
| siNC | siVAPA | 4.174 | 96.3% | 4 | 4 |

## Test family

- Due to the large variance, the following parametric tests are only used for comparison purposes. The sample size is based on the non-parametric tests listed above.
- One way ANOVA: Fixed effects, special, main effects and interactions, alpha error = 0.05

## Power calculations

- Performed with G*Power software, version 3.1.7 (*Faul et al., 2007*).
- ANOVA F test statistic and partial $\eta^2$ performed with R software, version 3.1.2 (*R Core Team 2015*).

| Groups | F test statistic | Partial η2 | Effect size f | A priori power | Total sample size |
|---|---|---|---|---|---|
| Optical density of DU145 cells transfected with siRNAs | $F_{(6, 24)} = 14.26$ | 0.7810 | 1.8884 | 82.73% | 16[1] |

[1] 60 total samples (5 per group) will be used based on the planned comparisons making the power >99.99%.

## Test family

- Two-tailed *t* test, difference between two independent means, Bonferroni correction: alpha error = 0.0167

## Power calculations

- Performed with G*Power software, version 3.1.7 (*Faul et al., 2007*).

| Cells | Group 1 | Group 2 | Effect size *d* | A priori power | Group 1 sample size | Group 2 sample size |
|---|---|---|---|---|---|---|
| DU145 | siNC | siPTEN | 5.115 | 92.9%[1] | 3[1] | 3[1] |
| | siNC | siCNO | 2.944 | 91.7% | 5 | 5 |
| | siNC | siVAPA | 4.174 | 97.6%[1] | 4[1] | 4[1] |

[1] 5 samples per group will be used making the power >99%.

## HCT116 cells
### Test family

- 2 tailed, Wilcoxon-Mann-Whitney test, Bonferroni's correction: alpha error = 0.00833

Power Calculations performed with G*Power software, version 3.1.7 (*Faul et al., 2007*).

| Cells | Group 1 | Group 2 | Effect size *d* | A priori power | Group 1 sample size | Group 2 sample size |
|---|---|---|---|---|---|---|
| HCT116 WT | siNC | siPTEN | 6.695 | 93.4% | 3 | 3 |
| | siNC | siCNO | 3.853 | 84.1% | 4 | 4 |
| | siNC | siVAPA | 5.463 | 80.5% | 3 | 3 |
| HCT116 Dicer Ex5 | siNC | siPTEN | 10.452 | 99.9% | 3 | 3 |
| | siNC | siCNO | 6.478 | 91.8% | 3 | 3 |
| | siNC | siVAPA | 4.128 | 89.1% | 4 | 4 |

## Test family

- Due to the large variance, the following parametric tests are only used for comparison purposes. The sample size is based on the non-parametric tests listed above.
- Two way ANOVA: Fixed effects, special, main effects and interactions, alpha error = 0.05

## Power calculations

- Performed with G*Power software, version 3.1.7 (*Faul et al., 2007*).
- ANOVA F test statistic and partial $\eta^2$ performed with R software, version 3.1.2 (*R Core Team 2015*).

| Groups | F test statistic | Partial η2 | Effect size f | A priori power | Total sample size |
|---|---|---|---|---|---|
| Optical density of HCT116 WT, and DICEREx5 cells transfected with siRNAs | F(3, 23) =14.08 | 0.7253 | 1.6249 | 81.69% | 12[1] |

[1]24 total samples (3 per group) will be used based on the planned comparisons making the power >99.99%.

## Test family

- Two-tailed *t* test, difference between two independent means, Bonferroni correction: alpha error = 0.00833

## Power calculations

- Performed with G*Power software, version 3.1.7 (*Faul et al., 2007*).

| Cells | Group 1 | Group 2 | Effect size *d* | A priori power | Group 1 sample size | Group 2 sample size |
|---|---|---|---|---|---|---|
| HCT116 WT | siNC | siPTEN | 6.695 | 96.3% | 3 | 3 |
| | siNC | siCNO | 3.853 | 87.9% | 3 | 3 |
| | siNC | siVAPA | 5.463 | 86.2% | 3 | 3 |
| Sensitivity Calculations | | | Detectable Effect size *d* | A priori power | Group 1 sample size | Group 2 sample size |
| HCT116 Dicer Ex5 | siNC | siPTEN | 3.495 | 80% | 4 | 4 |
| | siNC | siCNO | 3.495 | 80% | 4 | 4 |
| | siNC | siVAPA | 3.495 | 80% | 4 | 4 |

## Protocol 5

Summary of original AKT Activation data

- Note: data provided by original authors for Figure 5A.
  - We used the average band intensity for siNC since they were measured twice.

|  |  | pAkt/Total Akt | | |
|---|---|---|---|---|
| Cell Type | siRNA | 0 min | 5 min | 15 min |
| DU145 | siNC | 1 | 3.2 | 1.95 |
|  | siPTEN | 5 | 8.1 | 6.9 |
|  | siCNO | 0.8 | 4.1 | 3.4 |
|  | siVAPA | 2.1 | 10.9 | 6.6 |
| HCT116 WT | siNC | 1 | 2.35 | 2.45 |
|  | siPTEN | 6.3 | 9.5 | 9.5 |
|  | siCNO | 1.2 | 4.1 | 3 |
|  | siVAPA | 1.8 | 4.4 | 3.5 |
| HCT116 Dicer Ex5 | siNC | 1 | 5.15 | 2.25 |
|  | siPTEN | 5.3 | 14.7 | 7.2 |
|  | siCNO | 0.7 | 4.8 | 0.8 |
|  | siVAPA | 0.8 | 7 | 3.1 |

## DU145 cells

Note: The original data does not indicate the error associated with multiple biological replicates. To identify a suitable sample size, power calculations were performed using different levels of relative variance.

### Test family

- 2-Way ANOVA: Fixed effects, special, main effects and interactions, alpha error = 0.05 for DU145 cells

### Power calculations

- Performed with G*Power software, version 3.1.7 (*Faul et al., 2007*).
- ANOVA F test statistic and partial $\eta^2$ performed with R software, version 3.1.2 (*R Core Team 2015*).

| Groups | Variance estimate | F test statistic F(3,24) (siRNA main effect) | Partial $\eta^2$ | Effect size $f$ | A priori power | Total sample size |
|---|---|---|---|---|---|---|
| Akt activation in DU145 Cells transfected with siRNAs after 0, 5, and 15 min | 2% | 4598.79 | 0.9983 | 23.973 | 99.9% | 13 |
|  | 15% | 81.756 | 0.9109 | 3.1968 | 91.4% | 14 |
|  | 28% | 23.463 | 0.7457 | 1.7126 | 80.4% | 15 |
|  | 40% | 11.497 | 0.5897 | 1.1988 | 87.2% | 18 |

### Test family

- ANOVA F test statistic and planned contrasts with Bonferroni correction: alpha error = 0.01667

### Power calculations

- Performed with G*Power software, version 3.1.7 (*Faul et al., 2007*).

| Cells | Group 1 across time | Group 2 across time | Estimated variance | Effect size f | A priori power | Samples per group |
|---|---|---|---|---|---|---|
| DU145 | siNC | siPTEN | 2% | 18.533 | 92.0% | 2 |
| | | | 15% | 2.4710 | 98.6% | 2 |
| | | | 28% | 1.3238 | 82.6% | 2 |
| | | | 40% | 0.9266 | 83.7% | 2 |
| | siNC | siCNO | 2% | 2.8770 | 85.5% | 2 |
| | | | 15% | 0.3836 | 80.3% | 7 |
| | | | 28% | 0.2055 | 80.0% | 21 |
| | | | 40% | 0.1438 | 80.0% | 43 |
| | siNC | siVAPA | 2% | 17.997 | 91.1% | 2 |
| | | | 15% | 2.400 | 98.1% | 2 |
| | | | 28% | 1.2855 | 80.3% | 2 |
| | | | 40% | 0.8999 | 81.2% | 2 |

## HCT 116 cells

Note: The original data do not indicate the error associated with multiple biological replicates. To identify a suitable sample size, power calculations were performed using different levels of relative variance.

### Test family

- 3-Way ANOVA: Fixed effects, special, main effects and interactions, alpha error = 0.025 for HCT116[WT] and HCT116[DicerEx5] cells comparing AKT activation over time.

### Power calculations

- Performed with G*Power software, version 3.1.7 (*Faul et al., 2007*).
- ANOVA F test statistic and partial $\eta^2$ performed with R software, version 3.1.2 (*R Core Team 2015*).
  - For a given relative variance, 10,000 simulations were run and the F statistic and partial $\eta^2$ was calculated for each simulated data set.

| Groups | Variance Estimate | F test statistic $F_{(3,48)}$ (cell line, siRNA interaction) | Partial $\eta^2$ | Effect size f | A priori power | Total sample size |
|---|---|---|---|---|---|---|
| Akt activation in HCT116[WT] or HCT116[DicerEx5] cells transfected with siRNAs after 0, 5 and 15 min | 2% | 201.70 | 0.9173 | 3.3310 | 99.3% | 26 |
| | 15% | 4.7410 | 0.2162 | 0.5251 | 80.1% | 47 |
| | 28% | 2.1892 | 0.1128 | 0.3566 | 80.1% | 91 |
| | 40% | 1.6636 | 0.0878 | 0.3103 | 80.4% | 119 |

### Test family

- ANOVA F test statistic and planned contrasts with Bonferroni correction: alpha error = 0.01667 for each group of comparisons (cell type).

### Power calculations

- Performed with G*Power software, version 3.1.7 (*Faul et al., 2007*).

| Cells | Group 1 across time | Group 2 across time | | Effect size f | A priori power | Samples per group |
|---|---|---|---|---|---|---|
| HCT116<sup>WT</sup> | siNC | siPTEN | 2% | 18.322 | 99.9% | 2 |
| | | | 15% | 2.4429 | 99.9% | 2 |
| | | | 28% | 1.3087 | 99.9% | 2 |
| | | | 40% | 0.9161 | 99.9% | 2 |
| | siNC | siCNO | 2% | 2.3489 | 99.9% | 2 |
| | | | 15% | 0.3132 | 83.8% | 5 |
| | | | 28% | 0.1678 | 81.1% | 16 |
| | | | 40% | 0.1174 | 80.4% | 32 |
| | siNC | siVAPA | 2% | 3.6643 | 99.9% | 2 |
| | | | 15% | 0.4886 | 94.8% | 3 |
| | | | 28% | 0.2617 | 83.3% | 7 |
| | | | 40% | 0.1832 | 82.9% | 14 |

| Cells | Group 1 across time | Group 2 across time | | Effect size f | A priori power | Samples per group |
|---|---|---|---|---|---|---|
| HCT116<sup>DicerEx5</sup> | siNC | siPTEN | 2% | 17.664 | 99.9 | 2 |
| | | | 15% | 2.3552 | 99.9 | 2 |
| | | | 28% | 1.2617 | 99.9% | 2 |
| | | | 40% | 0.8832 | 99.9% | 2 |
| | Sensitivity calculation | | | Detectable effect size f | A priori power | Samples per group |
| | siNC | siCNO | 2% | 0.4971 | 80.0% | 2 |
| | | | 15% | 0.2999 | 80.0% | 5 |
| | | | 28% | 0.1742 | 80.0% | 16 |
| | | | 40% | 0.1170 | 80.0% | 32 |
| | Sensitivity calculation | | | Detectable effect size f | A priori power | Samples per group |
| | siNC | siVAPA | 2% | 0.4971 | 80.0% | 2 |
| | | | 15% | 0.2999 | 80.0% | 5 |
| | | | 28% | 0.1742 | 80.0% | 16 |
| | | | 40% | 0.1170 | 80.0% | 32 |

In order to produce quantitative replication data, we will run the experiment seven times. Each time we will quantify band intensity. We will determine the standard deviation of band intensity across the biological replicates and combine this with the reported value from the original study to simulate the original effect size. We will use this simulated effect size to determine the number of replicates necessary to reach a power of at least 80%. We will then perform additional replicates, if required, to ensure that the experiment has more than 80% power to detect the original effect.

## Acknowledgements

The Reproducibility Project: Cancer Biology core team would like to thank Yvonne Tay and Pier Paolo Pandolfi for generously sharing critical information as well as reagents to ensure the fidelity and quality of this replication attempt. We thank Dr. Bert Vogelstein for directing us to the appropriate facility to obtain HCT116 DICER mutant cells. We thank Courtney Soderberg at the Center for Open Science for assistance with statistical analyses. We would also like to thank the following companies for generously donating reagents to the Reproducibility Project: Cancer Biology; American Type and Tissue Collection (ATCC), Applied Biological Materials, BioLegend, Charles River

Laboratories, Corning Incorporated, DDC Medical, EMD Millipore, Harlan Laboratories, LI-COR Biosciences, Mirus Bio, Novus Biologicals, Sigma-Aldrich, and System Biosciences (SBI).

## Additional information

### Group author details

Reproducibility Project: Cancer Biology

Elizabeth Iorns: Science Exchange, Palo Alto, United States; William Gunn: Mendeley, London, United Kingdom; Fraser Tan: Science Exchange, Palo Alto, United States; Joelle Lomax: Science Exchange, Palo Alto, United States; Nicole Perfito: Science Exchange, Palo Alto, United States; Timothy Errington: Center for Open Science, Charlottesville, United States

### Competing interests

MP: The Ohio State University Pharmacoanalytical Shared Resource is a Science Exchange associated laboratory. CC: The Ohio State University Pharmacoanalytical Shared Resource is a Science Exchange associated laboratory. HW: The Ohio State University Pharmacoanalytical Shared Resource is a Science Exchange associated laboratory. RP:CB: EI, FT, NP, and JL are employed and hold shares in Science Exchange, Inc. The other authors declare that no competing interests exist.

### Funding

| Funder | Author |
| --- | --- |
| Laura and John Arnold Foundation | Reproducibility Project: Cancer Biology |

The funders had no role in study design, data collection and interpretation, or the decision to submit the work for publication.

### Author contributions

MP, CC, HW, MC, Drafting or revising the article; RP:CB, Conception and design, Drafting or revising the article

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
