## [Decision Letter]

Thank you for submitting your work entitled "Registered report: Coding-independent regulation of the tumor suppressor PTEN by competing endogenous mRNAs" for consideration by *eLife*. Your article has been reviewed by four peer reviewers, and the evaluation has been overseen by a Reviewing Editor and Tony Hunter as the Senior Editor. One of the four reviewers, Klaus Rajewsky (Reviewer 4), has agreed to reveal his identity.

Your Registered report has been reviewed by four expert referees. As you will see, all are quite positive about the proposed work. Please address the very minor points raised by the reviewers before uploading your final files but consider the Report to be In Press.

Reviewer #1:

This Registered report describes the proposed replication plan of key experiments from "Coding-Independent Regulation of the Tumor Suppressor PTEN by Competing Endogenous mRNAs" by Tay and colleagues, published in Cell in 2011 (Tay et al., 2011).

For all protocols, the authors propose use ANOVA to analyze the data. Please check for outliers and make sure that the data do not violate the assumptions of the ANOVA: normality and homoscedasticity. If the data do not fit the assumptions well enough, try to find a data transformation that makes them fit. If this doesn't work, suggest/apply a nonparametric counterpart of ANOVA.

Reviewer #2:

The authors of this report propose to replicate experiments within Coding-independent Regulation of the Tumor Suppressor PTEN by Competing Endogenous mRNAs, by Tay et al., 2011. This study reported a set of genes (NCOA7, BCL11B, SERINC1, ZNF460, NUDT13, DTWD2, and VAPA) regulating the expression of the tumor suppressor PTEN by acting as competing endogenous RNAs (ceRNAs). The authors describe the following as the essential results of Tay et al., 2011: 1.) When DU145 cells are transfected with a luciferase construct containing the PTEN 3′UTR and siRNAs against each of the putative ceRNAs, luciferase activity decreases in comparison to transfections with the construct and a control siRNA. 2.) When the same cells are transfected with a luciferase construct containing the PTEN 3′UTR and a construct containing the 3′UTR of one of the ceRNAs, luciferase activity increases in comparison to when transfected with the construct and a control construct. 3.) When HCT WT cells are transfected with siRNAs against each of the identified ceRNAs, PTEN expression as measured by protein blot decreases in comparison to transfections with a control siRNA. When this experiment is repeated in HCT Dicer^Ex5^, which is impaired in production of miRNA levels, the reduction of PTEN upon ceRNA knockdown is abrogated, supporting the idea that the response to modulating the ceRNAs is miRNA dependent. 4.) When DU125 and HCT WT cells are transected with an siRNA against PTEN or one of two ceRNAs (CNO a VAPA), cell proliferation increases in comparison to when transfected with a control siRNA. When this experiment is repeated in HCT Dicer^Ex5^, the increased proliferation upon knockdown of either of the two ceRNAs, but not PTEN, is reduced. 5.) When DU125 and HCT WT cells are transected with an siRNA against PTEN or one of two ceRNAs (CNO a VAPA) and serum starved, phosphorylation of Akt increases after restimulation, in comparison to when transfected with a control siRNA. When this experiment is repeated in HCT Dicer^Ex5^, the increase in Akt phosphorylation upon knockdown of either of the two ceRNAs, but not PTEN, is abrogated.

Considered questions:

1) Do the experiments chosen embody the main conclusions drawn from the original article?

These experiments embody the main conclusions. Protocols 1 and 2 are designed to demonstrate that each ceRNA positively regulates PTEN protein levels through the 3′ UTR of both the ceRNA and PTEN transcript. Protocol 3 is designed to demonstrate that this effect is dependent on miRNAs. Protocol 4 is designed to demonstrate that loss of PTEN or its ceRNAs increases cell proliferation, and Protocol 5 is designed to demonstrate that loss of PTEN or its ceRNAs increases Akt phosphorylation, which is a proliferation signal.

2) Do the authors accurately summarize the literature, especially with respect to other direct replications?

Yes.

3) Are the proposed experiments appropriately designed?

The original experiments corresponding to each of the five protocols had only a single siRNA or UTR control. If the authors had the latitude to add more controls, the results would be more robust, although this would go beyond the scope of simply repeating the published experiments. In Protocol 3 and 5 the protein blots could be performed loading a dilution series of total protein (e.g., 5 µg, 2 µg, 1 µg) from the control sample, to ensure that quantitation is in the linear range and not confounded by overexposure (a concern of the original authors).

3) Are the proposed statistical analyses rigorous and appropriate?

Yes.

4) What can the replication team do to maximize the quality of the replication?

The team has done a thorough job in designing this attempted replication.

Reviewer #3:

The authors present a clear, well-controlled plan for this replication study. They have also included comments and experimental details provided by the original authors. They should address the minor comments listed below before this manuscript can be accepted for publication.

Comments for the authors:

Paragraph one, Introduction – cognante should be cognate.

Paragraph three, Introduction – CNOTL6 should be CNOT6L.

Paragraph eight, Introduction – The Poliseno group should be The Pandolfi group.

Protocol 1, “Materials and Reagents” table (and all other mentions of the TaqMan probes) – The original product numbers are specified in the extreme left column. For example, the PTEN TaqMan probe used is Hs02621230_s1.

Protocol 5, “Materials and Reagents” table – The P-Akt antibody should be 9271 (Cell signalling). This is for P-Akt Ser473, which is what was examined in the original paper. Cat number 9275 is for the P-Akt Thr308 antibody.

Reviewer #4:

We have carefully checked the proposal with respect to the 5 criteria specified in the reviewers' guidelines and found the proposal just perfect. Of course nowadays one would like to see the Pandolfi experiments controlled by CRISPR/Cas mutagenesis, but this is apparently not part of the present replication program.

---

## [Author Response]

Reviewer #1:

For all protocols, the authors propose use ANOVA to analyze the data. Please check for outliers and make sure that the data do not violate the assumptions of the ANOVA: normality and homoscedasticity. If the data do not fit the assumptions well enough, try to find a data transformation that makes them fit. If this doesn't work, suggest/apply a nonparametric counterpart of ANOVA.

We appreciate the point that the reviewer has brought up. We have added the following statement to the analysis sections where appropriate.

“Note: At the time of analysis we will perform the Shapiro-Wilk test and generate a quantile-quantile plot to assess the normality of the data. We will also perform Levene’s test to assess homoscedasticity. If the data appears skewed we will perform the appropriate transformation in order to proceed with the proposed statistical analysis. If this is not possible we will perform the equivalent non-parametric Wilcoxon-Mann-Whitney test.”

Reviewer #2:

The authors of this report propose to replicate experiments within Coding-independent Regulation of the Tumor Suppressor PTEN by Competing Endogenous mRNAs, by Tay et al., 2011. This study reported a set of genes (NCOA7, BCL11B, SERINC1, ZNF460, NUDT13, DTWD2, and VAPA) regulating the expression of the tumor suppressor PTEN by acting as competing endogenous RNAs (ceRNAs). The authors describe the following as the essential results of Tay et al., 2011: 1.) When DU145 cells are transfected with a luciferase construct containing the PTEN 3′UTR and siRNAs against each of the putative ceRNAs, luciferase activity decreases in comparison to transfections with the construct and a control siRNA. 2.) When the same cells are transfected with a luciferase construct containing the PTEN 3′UTR and a construct containing the 3′UTR of one of the ceRNAs, luciferase activity increases in comparison to when transfected with the construct and a control construct. 3.) When HCT WT cells are transfected with siRNAs against each of the identified ceRNAs, PTEN expression as measured by protein blot decreases in comparison to transfections with a control siRNA. When this experiment is repeated in HCT Dicer^Ex5^, which is impaired in production of miRNA levels, the reduction of PTEN upon ceRNA knockdown is abrogated, supporting the idea that the response to modulating the ceRNAs is miRNA dependent. 4.) When DU125 and HCT WT cells are transected with an siRNA against PTEN or one of two ceRNAs (CNO a VAPA), cell proliferation increases in comparison to when transfected with a control siRNA. When this experiment is repeated in HCT Dicer^Ex5^, the increased proliferation upon knockdown of either of the two ceRNAs, but not PTEN, is reduced. 5.) When DU125 and HCT WT cells are transected with an siRNA against PTEN or one of two ceRNAs (CNO a VAPA) and serum starved, phosphorylation of Akt increases after restimulation, in comparison to when transfected with a control siRNA. When this experiment is repeated in HCT Dicer^Ex5^, the increase in Akt phosphorylation upon knockdown of either of the two ceRNAs, but not PTEN, is abrogated.

Considered questions:

1) Do the experiments chosen embody the main conclusions drawn from the original article?These experiments embody the main conclusions. Protocols 1 and 2 are designed to demonstrate that each ceRNA positively regulates PTEN protein levels through the 3′ UTR of both the ceRNA and PTEN transcript. Protocol 3 is designed to demonstrate that this effect is dependent on miRNAs. Protocol 4 is designed to demonstrate that loss of PTEN or its ceRNAs increases cell proliferation, and Protocol 5 is designed to demonstrate that loss of PTEN or its ceRNAs increases Akt phosphorylation, which is a proliferation signal.2) Do the authors accurately summarize the literature, especially with respect to other direct replications?Yes.

3) Are the proposed experiments appropriately designed?The original experiments corresponding to each of the five protocols had only a single siRNA or UTR control. If the authors had the latitude to add more controls, the results would be more robust, although this would go beyond the scope of simply repeating the published experiments. In Protocol 3 and 5 the protein blots could be performed loading a dilution series of total protein (e.g., 5 µg, 2 µg, 1 µg) from the control sample, to ensure that quantitation is in the linear range and not confounded by overexposure (a concern of the original authors).

We agree with the reviewer that there can be much performed outside of what would be considered a direct replication and that these questions should be answered outside of this experimental setup. As for the Western blotting protocols, multiple exposures will be taken at various times to minimize the risk of overexposure with all images made publically available.

Reviewer #3:Paragraph one, Introduction – cognante should be cognate

This has been corrected in the revised manuscript.

Paragraph three, Introduction – CNOTL6 should be CNOT6L

This has been corrected in the revised manuscript.

Paragraph eight, Introduction – The Poliseno group should be The Pandolfi group

This has been corrected in the revised manuscript.

Protocol 1, “Materials and Reagents” table (and all other mentions of the TaqMan probes) – The original product numbers are specified in the extreme left column. For example, the PTEN TaqMan probe used is Hs02621230_s1.

We have moved the product numbers for each TaqMan probe to the appropriate fourth column and removed the comment that the original product number was not specified.

Protocol 5, “Materials and Reagents” table – The P-Akt antibody should be 9271 (Cell signalling). This is for P-Akt Ser473, which is what was examined in the original paper. Cat number 9275 is for the P-Akt Thr308 antibody.

Thank you for this comment. We confirmed with the original authors that the catalog number 9271 (for the P-Akt Ser473) should be used and have corrected this in the revised manuscript.

Reviewer #4:We have carefully checked the proposal with respect to the 5 criteria specified in the reviewers' guidelines and found the proposal just perfect. Of course nowadays one would like to see the Pandolfi experiments controlled by CRISPR/Cas mutagenesis, but this is apparently not part of the present replication program.

We appreciate the reviewers’ note and agree that such exploratory analyses would be appropriate for future replication attempts.